# Photosynthate distribution determines spatial patterns in the rhizosphere microbiota of the maize root system

Sina R. Schultes[1], Lioba Rüger[2], Daniela Niedeggen[2], Jule Freudenthal[2], Katharina Frindte[1], Maximilian F. Becker[1], Ralf Metzner[3], Daniel Pflugfelder[3], Antonia Chlubek[3], Carsten Hinz[3], Dagmar van Dusschoten[3], Sara L. Bauke[4], Michael Bonkowski[2], Michelle Watt[3,5], Robert Koller[3] ✉ & Claudia Knief[1] ✉

The spatial variation and underlying mechanisms of pattern formation in the rhizosphere microbiome are not well understood. We demonstrate that specific patterns in the distribution of recently fixed carbon within the plant root system influence the spatial organization of the rhizosphere microbiota. Non-invasive analysis of carbon allocation in the maize root system by [11]C tracer-based positron emission tomography combined with magnetic resonance imaging reveals high spatial heterogeneity with highest [11]C-signal accumulations at root tips and differences between root types. Strong correlations exist between root internal carbon allocation and rhizodeposition as evident from [13]$CO_2$ labeling. These patterns are reflected in the bacterial, fungal and protistan community structure in rhizosphere soil with differences depending on root structure and related spatial heterogeneities in carbon allocation. Especially the active consumers of [13]C-labeled rhizodeposits are responsive to photosynthate distribution with differences in [13]C-labeling according to their spatial localization within the root system. Thus, root photosynthate allocation supports distinct habitats in the plant root system and is a key determinant of microbial food web development, evident from [13]C-labeling of diverse bacterial and protistan predators, especially at root bases, resulting in characteristic spatiotemporal patterns in the rhizosphere microbiome.

The assembly of the rhizosphere microbiome is the result of complex interactions between the plant and microorganisms. It is further affected by abiotic and biotic factors such as soil properties and interactions between microorganisms in the rhizosphere. It is suggested that microbiome enrichment by the plant host is, for a large part, mediated by rhizodeposition of various compounds into the rhizosphere[1,2]. Plants exude a substantial amount of carbon into the rhizosphere[3], and different exudate compounds have been reported to attract specific microbial consumers[4–6]. These microbial consumers of rhizodeposits have been identified using [13]$CO_2$ labeling followed by DNA-based stable isotope probing (DNA-SIP)[7].

[1]Institute for Crop Science and Resource Conservation (INRES), Molecular Biology of the Rhizosphere, University of Bonn, Bonn, Germany. [2]Institute of Zoology, Terrestrial Ecology, Cluster of Excellence on Plant Sciences (CEPLAS), University of Cologne, Cologne, Germany. [3]Institute of Bio- and Geosciences, IBG 2: Plant Sciences, Forschungszentrum Jülich GmbH, Jülich, Germany. [4]Institute for Crop Science and Resource Conservation (INRES), Soil Sciences, University of Bonn, Bonn, Germany. [5]Present address: Adrienne Clarke Chair of Botany, School of BioSciences, Faculty of Science, University of Melbourne, Melbourne, VIC, Australia. ✉e-mail: r.koller@fz-juelich.de; knief@uni-bonn.de

The root system is mostly sampled as a whole, but no studies have yet systematically unraveled how spatially heterogeneous carbon allocation and rhizodeposition within the root system affect microbiome establishment and lead to spatial patterns in the rhizosphere microbiome. Rhizodeposition is known to be spatially heterogeneous within the root system[8,9]. It varies along the root axis[8,10] and root exudates have been reported to differ between root types[11,12]. Similarly, evidence exists for heterogeneities in the rhizosphere microbiome within plant root systems[13–16], and differences have likewise been reported for microbial communities along the root axis[10] and between root types[11,12]. These heterogeneities in rhizodeposition and microbial community composition suggest the presence of consistent, small-scale selection mechanisms in the rhizosphere, likely driven by substrate preferences of the rhizosphere microbiota[2]. However, apart from the notion that a specific bacterial reporter strain profited primarily from photosynthate distribution at the root tips[17], dependencies within the root system remain largely unresolved at the small scale. It is unclear to what extent spatial variation in the rhizosphere microbiome is related to spatial patterns in carbon allocation within the root system and to rhizodeposition. This leads to the question where in the root system specific microbial taxa are particularly supported by rhizodeposits. Such small-scale dependencies will guide microbial community establishment and processes in the rhizosphere. This in turn has possible implications for the whole food-web that is fueled by rhizodeposits and possibly for plant performance, considering that the rhizosphere microbiome includes taxa with potential benefits for the plant[1,18].

Here, we assess congruencies between spatial patterns in root carbon allocation, rhizodeposition and the rhizosphere microbiota within the maize root system. Maize has a complex root system consisting of different root types with seed-borne (primary and seminal) roots and shoot-borne (crown) roots that develop sequentially over time from different nodes (Supplementary Fig. 1)[19,20]. Root carbon allocation was studied by $^{11}CO_2$ pulse labeling of plants coupled with positron emission tomography (PET) and magnetic resonance imaging (MRI). The short-lived carbon radioisotope $^{11}C$ ($t_{1/2} = 20.4$ min) allows for tracing recently fixed carbon in vivo and enables detailed time series of $^{11}C$ tracer allocation processes in the root system[21–23]. Using image-guided sampling for destructive sample collection, data on photosynthate allocation were integrated with microbial data from community compositional analysis (study I). In a second study, this approach was complemented by photosynthate labeling using $^{13}CO_2$ as stable isotopic tracer to track the transfer of carbon from the root into the rhizosphere and its microbiota, followed by DNA-based stable isotope probing (SIP) to identify key microbial consumers of rhizodeposits (Fig. 1). We analyzed the prokaryotic, fungal and cercozoan consumers, the latter representing a major group of soil protists[24]. We hypothesize that (I) photosynthates are heterogeneously distributed in the root system, especially along the longitudinal root axis and between root types. (II) Root-internal heterogeneity in carbon allocation extends into the rhizosphere with a largely congruent pattern. (III) The rhizosphere microbiota is expected to develop specific spatial patterns in the rhizosphere in response to carbon allocation. (IV) Microbial taxa are particularly supported by rhizodeposits related to their localization within the root system. This would make photosynthate distribution a central driver of microbial small-scale heterogeneity in the rhizosphere of a root system.

## Results

### PET-MRI imaging reveals highly heterogeneous distribution of recently fixed carbon

In the two experimental studies, the root systems of all plants were regularly imaged by PET-MRI to non-invasively collect data on photosynthate allocation and root growth. $^{11}C$-PET in combination with MRI measurements revealed that recently fixed carbon was heterogeneously distributed in the plant root system over time (Fig. 2). Nevertheless, consistent patterns were observed among the $^{11}CO_2$ labeled plants in study I (Supplementary Fig. 2) and study II (Supplementary Fig. 3).

We evaluated the spatial patterns in $^{11}C$-photosynthate allocation into the root system at three different plant developmental stages in study I. After 6 days of plant development, no $^{11}C$ tracer was detected in the root system, indicating that recently fixed carbon was not yet allocated into the roots (Fig. 2A). In contrast, intensive and heterogeneous belowground allocation of carbon was observed in plants of 13 and 20 days of age, with primary, seminal and crown root tips exhibiting high levels of $^{11}C$ tracer signal. Tips of the most recently emerged crown roots held particularly strong accumulations of the tracer. On day 13, the tips of crown roots of the second node were rapidly supplied with recently fixed carbon and showed the highest $^{11}C$ signal intensity. On day 20, the newly emerging crown roots of the third and fourth node showed particularly high signal intensity, while intensity in crown roots of the second node had declined as the roots matured. Lateral root tips also showed photosynthate accumulations, but this was more difficult to resolve due to their high number and overlapping tracer signals.

Belowground $^{11}C$-photosynthate allocation into the root system occurred very rapidly (Fig. 2B, Vid. S1), with the first photosynthates being detected in the primary root of 13-day-old plants at around 10–15 min after the start of the pulse labeling (Vid. S1). The first accumulations of labeled photosynthates at root tips were detected in the youngest crown roots (Fig. 2B, Vid. S1). Once established, these accumulations at root tips generally persisted over the total measurement time, whereas the tracer signal disappeared at the root bases of some roots over time including the primary root, which had received the tracer first.

### Root photosynthate allocation as main driver of rhizodeposition

To relate carbon allocation inside of the roots to carbon allocation in the rhizosphere, the $^{11}C$-PET analysis was complemented by stable isotope labeling of shoots with $^{13}CO_2$ and EA-IRMS measurements of root tissue and rhizosphere samples from root tips (2 cm length) and bases (upper 10 cm of a root) after destructive sampling in study II. The $^{11}CO_2$ labeling pulse was again 6 min long, whereas plants were exposed to 12-h $^{13}CO_2$ pulses daily for the preceding 6 days to achieve good label incorporation into the microbiota for subsequent DNA-SIP.

As observed in $^{11}C$-PET imaging, the mass fraction of $^{13}C$ in the root tissue and rhizosphere soil was noticeably higher in most root tips compared to their bases, except for the root tissue from the youngest crown roots (Fig. 3A, B). Differences were also evident between different root types, whereby the root tissue and rhizosphere soil at the root bases showed a consistent increase in mass fraction of $^{13}C$ from the primary root over the seminal roots to the different generations of crown roots. A comparison of the mass fraction of $^{13}C$ between root tissues and corresponding rhizosphere samples revealed consistently higher values in root tissue, along with a significantly positive Pearson correlation ($r = 0.77$, $p < 0.001$) (Fig. 3C). When this was analyzed separately for root tips and bases, it became evident that this correlation was primarily driven by a very strong correlation in samples from the root bases ($r = 0.96$, $p < 0.001$; Fig. 3D), whereas correlation at the root tips was clearly weaker and not significant anymore ($r = 0.44$, $p = 0.054$). The absence of a significant correlation at the root tips resulted from differences in carbon allocation. In the root tips, the mass fraction of $^{13}C$ was higher for younger roots compared to older ones, being highest in the first generation of crown roots. In contrast, mass fractions of $^{13}C$ in the rhizosphere soil were less distinct and in case of the primary and seminal roots more heterogeneous. The decline in $^{13}C$ in the root tips of the youngest (third) generation of crown roots compared to the first and second generation appeared to be in contrast to the observed highest $^{11}C$ signals in these youngest roots. This can be explained by the fastest growth of these roots, along with the lag period between the last $^{13}CO_2$ pulse and sample collection, which was required for $^{11}C$-PET and MRI

# Study I

**Day 6, 13 & 20: PET/MRI imaging of $^{11}$C allocation**

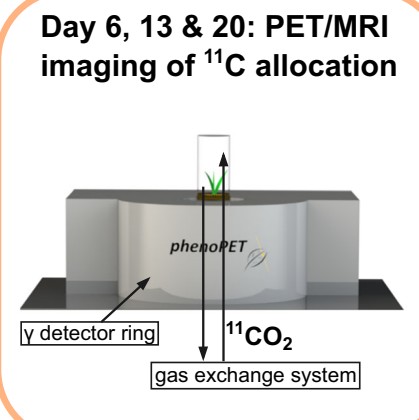

γ detector ring | $^{11}CO_2$ | gas exchange system

**Day 21: Image-guided sampling of rhizosphere**

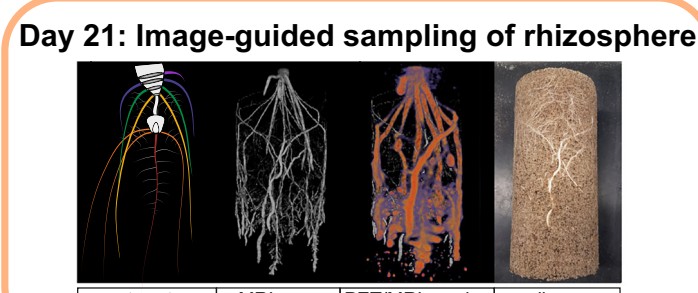

| root system | MRI scan | PET/MRI overlay | soil core |

**16S, ITS and 18S qPCR and amplicon sequencing**

# Study II

**Day 14 & 21: PET/MRI imaging of $^{11}$C allocation**

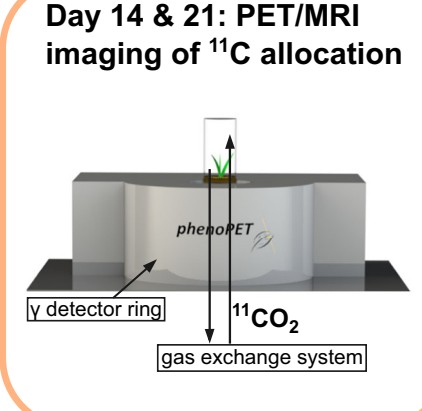

γ detector ring | $^{11}CO_2$ | gas exchange system

**Day 15-20: $^{13}CO_2$ labeling (12h / day)**

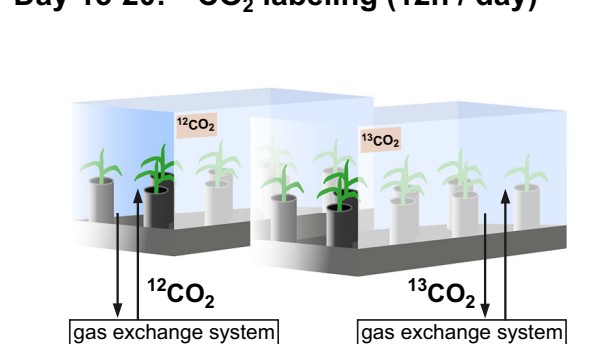

$^{12}CO_2$ | gas exchange system | $^{12}CO_2$

$^{13}CO_2$ | gas exchange system | $^{13}CO_2$

**Day 22: Image-guided sampling of roots and rhizosphere**

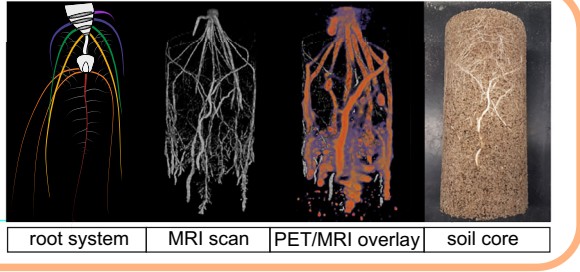

| root system | MRI scan | PET/MRI overlay | soil core |

**Quantification of $^{13}$C in root and rhizosphere samples by EA-IRMS**

**Identification of $^{13}$C consumers in the rhizosphere by DNA-SIP and 16S, ITS and 18S amplicon sequencing**

**Fig. 1 | Conceptual study design.** In Study I we investigated temporal changes in photosynthate allocation and the effect of small-scale spatial heterogeneity in photosynthate allocation on the prokaryotic, fungal and cercozoan rhizosphere communities. PET-MRI imaging (orange boxes) enabled us to define root regions with distinct photosynthate levels (low, medium, high) and to consider root architecture at high spatial resolution for identifying root types and sections during sampling. Study II combines this approach with $^{13}CO_2$ labeling (blue box) to specifically identify photosynthate consumers and investigate their response to spatially heterogeneous photosynthate distribution. Analyses upon destructive sampling (gray boxes) included elemental analysis coupled to isotope ratio mass spectrometry (EA-IRMS) to obtain quantitative data on photosynthate contents of root and rhizosphere samples and DNA-based microbial community compositional analyses.

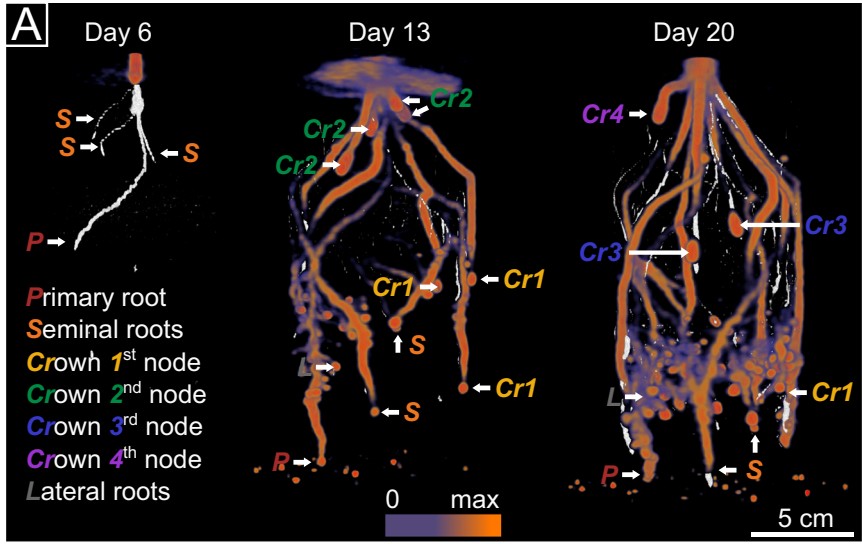

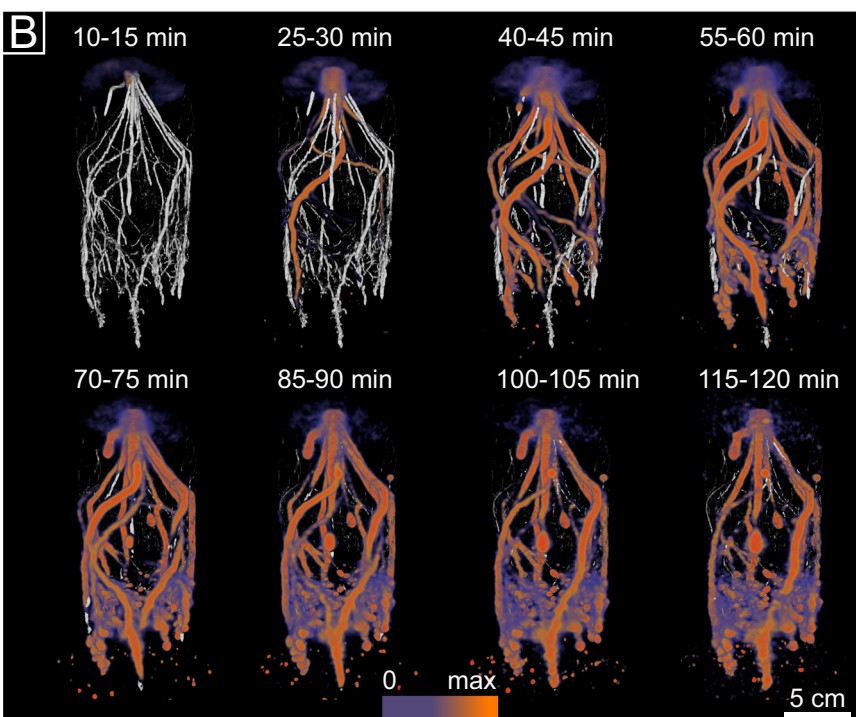

**Fig. 2 | Allocation of recently fixed carbon in roots visualized by co-registered PET-MRI scans.** The same maize plants were imaged in study I at three time points by PET (colored) and MRI (gray) over a period of 120 min. **A** Co-registered PET-MRI scans were obtained for 6-, 13-, and 20-day-old plants at 85–90 min after labeling start. Prominently visible representatives of the different root types are marked with letters: P primary root, S seminal roots, Cr crown roots of nodes 1-4, L lateral roots. On day 6, the root system consisted of the primary root and three seminal roots. On day 20, around 10–12 crown roots from up to four different consecutive nodes had developed in all plants. **B** Belowground allocation of $^{11}$C-labeled photosynthates in a 20-day-old plant over the measurement period of 2 h. Images are integrals calculated over 5 min of measurement. A time-lapse video showing $^{11}$C-photosynthate allocation is available as Supplementary Movie 1.

imaging. This resulted in further biomass formation at the root tips with unlabeled photosynthates. However, this lag period after $^{13}$C pulse labeling does not explain the absence of correlation at the root tips, because the exclusion of the youngest, fastest growing crown root tips from the correlative analysis did not result in significant findings. The mass fraction of $^{13}$C in unlabeled controls matched closely the isotope's natural abundance of -1.1% (Supplementary Fig. 4).

### Root architectural factors and photosynthate level determine microbial community composition

To investigate how photosynthate allocation, root type and root section affect the composition of the rhizosphere microbiota, we sampled root tips and bases from all root types and defined for these samples three levels of photosynthate allocation based on $^{11}$C signal intensities (Supplementary Fig. 1). We distinguished between root tips with high signal intensity, root tips with medium signal intensity and basal root sections with low signal intensity. The community compositional analysis targeted prokaryotes, fungi and Cercozoa. In study I, the communities were directly analyzed by amplicon sequencing, whereas the focus in study II was on the communities that incorporated $^{13}$C-labeled photosynthates to more closely link carbon allocation patterns to the benefiting microbial taxa. $^{13}$C-incorporating taxa were represented by $^{13}$C-labeled DNA obtained from a heavy fraction upon density gradient centrifugation of the DNA. In both experimental

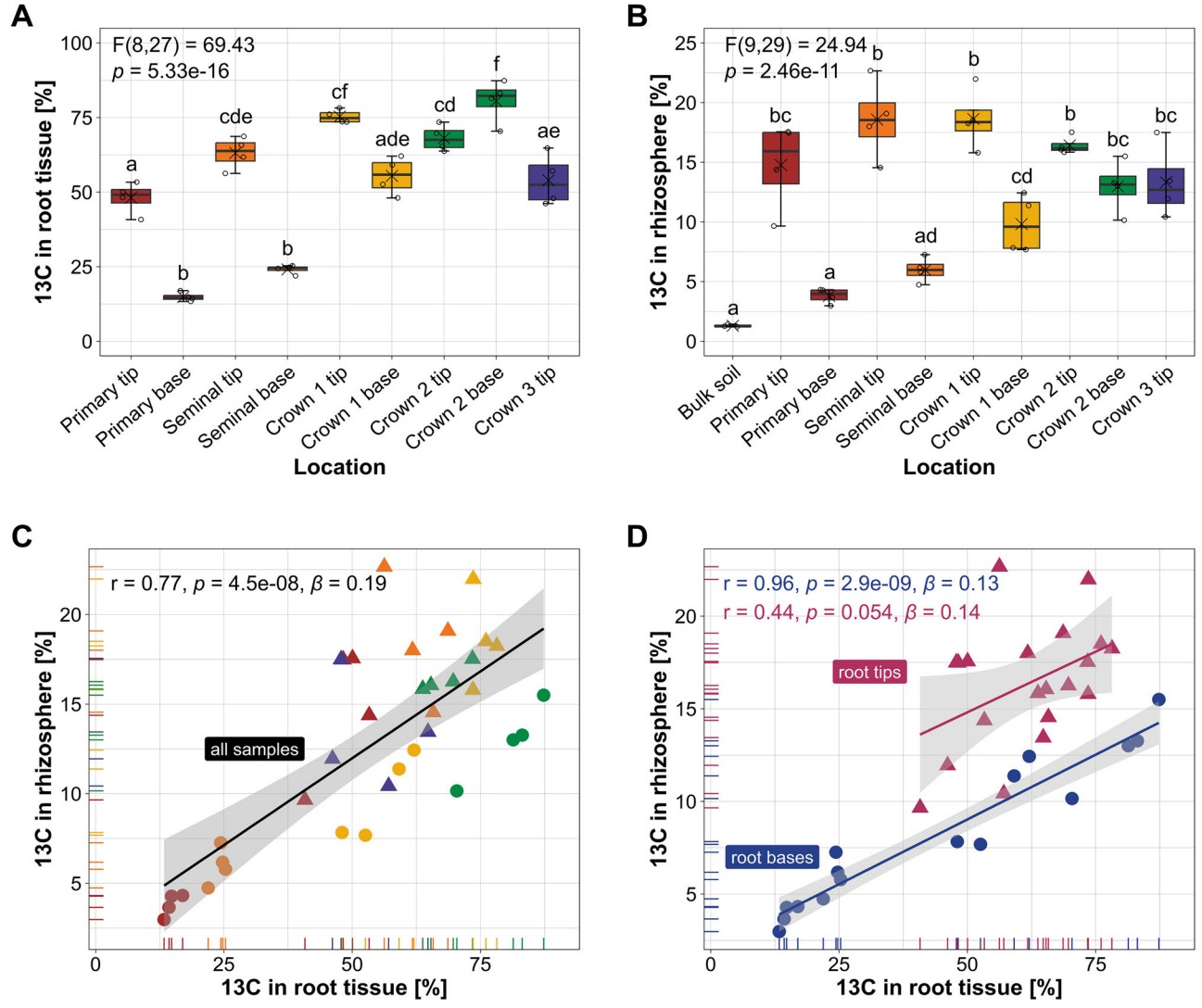

**Fig. 3 | $^{13}$C tracer allocation in different root types and root sections and their associated rhizosphere soil samples.** The mass fraction of $^{13}$C in percent in **A** root tissue and **B** rhizosphere soil samples as measured by EA-IRMS and visualized in box plots across different locations in the root system. Boxes span from the first to the third quartiles, the line inside each box represents the median; x represents the mean and the whiskers extend to the last data point within 1.5 times the inter quartile range. Data points outside of whiskers represent outliers. Significance of differences was tested by one-way ANOVA (two-sided), and distinct lowercase letters indicate significant differences between groups ($n = 4$ biological replicates) as tested by Tukey HSD post hoc tests ($p < 0.05$). Scatter plots illustrate the relationship (Pearson correlation coefficient r with significance ($p$)) between mass fraction $^{13}$C in root tissue and the rhizosphere for **C** all samples colored by root type ($n = 36$ biological replicates) and **D** for root tip ($n = 20$) and root base samples ($n = 16$) separately. $\beta$ indicates the slope. A linear regression model was fitted (solid black line), and the shaded area represents the 95% confidence interval around the best fit (based on the standard error of the estimate). Source data are provided as a Source Data file.

studies, the composition of the prokaryotic, fungal and cercozoan communities differed according to root type, root section and the $^{13}$C signal categories, which was confirmed by permutational analysis of variance (PERMANOVA) (Table 1, Supplementary Table 1). The statistical model "root type * $^{13}$C level" had the best explanatory power in study I. Thereby, root section could be excluded from the model, because the other two factors covered the variation by root section. When root section was introduced as first term in the model, significant variation was assigned to this factor and $^{13}$C level could be eliminated.

The non-metric multidimensional scaling (NMDS) plots revealed clear clustering according to root type, especially for bacteria (Fig. 4A, Supplementary Fig. 5). Even the different generations of crown roots formed discernable clusters. This patterning became more pronounced in the microbiota represented by the $^{13}$C-labeled DNA fraction in study II, especially in case of fungi, whereas the clustering of Cercozoa related to root type remained less distinct. A sample grouping

related to root type was likewise seen in the PERMANOVA results with $R^2$ values ranging in study I from 0.13 to 0.20 and in study II from 0.14 to 0.26 plus, in case of bacteria and fungi, an increased interaction term in study II (Table 1). Sample clustering by root type in the NMDS plots even reflected the chronology of root emergence from the primary root over the seminal roots towards the first, second, third and fourth generation of crown roots. This successive clustering in the NMDS plot was also particularly evident for the bacteria in study II. Further, it was very well reflected in $R^2$ values of PERMANOVA post-hoc comparisons between all individual root types, though the rather low sample number per root type along with the correction for multiple comparisons did not support the significance of most $R^2$ values (Supplementary Fig. 6). Moreover, the successiveness across root types was seen in the analysis of differentially abundant taxa (Supplementary Fig. 6, Supplementary Data 1, Supplementary Table 2). This revealed a couple of abundant genera with monotonic changes in relative abundance in the rhizosphere from older towards younger roots. For

## Table 1 | Influence of root type and photosynthate level on microbial community composition

| Study | PERMANOVA model | Factor | Bacteria | Fungi | Cercozoa |
|---|---|---|---|---|---|
| **Study I:** Total community | Root type * [11]C level | Root type | 0.20*** | 0.13*** | 0.18*** |
| | | [11]C level (categorical) | 0.07*** | 0.08*** | 0.09*** |
| | | Interaction | 0.07*** | 0.08** | 0.04 |
| | | Residuals | 0.66 | 0.71 | 0.69 |
| **Study II:** [13]C-heavy fraction community | Root type * [11]C level | Root type | 0.26*** | 0.14* | 0.17** |
| | | [11]C level (categorical) | 0.34*** | 0.35*** | 0.47*** |
| | | Interaction | 0.13*** | 0.14** | 0.05 |
| | | Residuals | 0.27 | 0.37 | 0.31 |
| | Root type * [13]C level | Root type | 0.26*** | 0.14 | 0.17* |
| | | [13]C level (numeric) | 0.31*** | 0.34*** | 0.37*** |
| | | Interaction | 0.11** | 0.10 | 0.05 |
| | | Residuals | 0.32 | 0.42 | 0.41 |

Influence of both factors as well as their interaction on total community composition (study I) and the photosynthate consumer community, reflected by the [13]C-heavy fraction of the DNA-SIP analysis (study II), was assessed by PERMANOVA based on 999 permutations. $R^2$ values are given in the table, detailed test statistics in Supplementary Table 1. Significance code: *** indicates $p \leq 0.001$, **$p \leq 0.01$ and *$p \leq 0.05$.

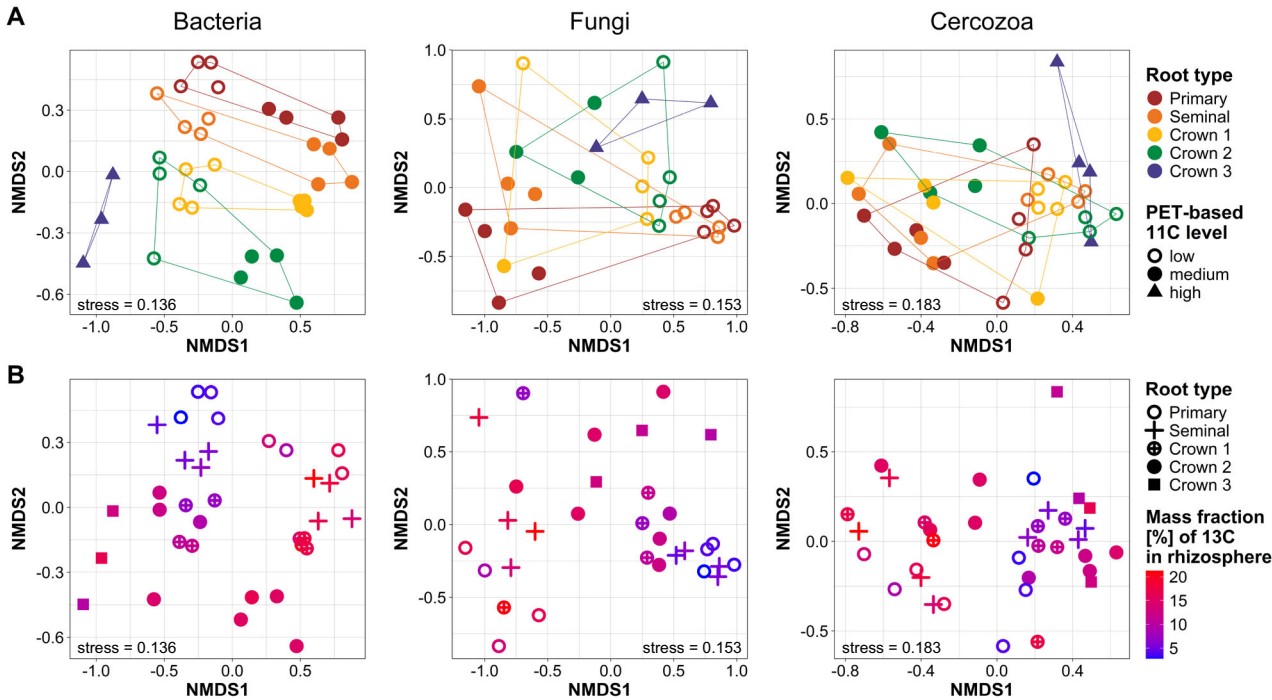

**Fig. 4 | Variation in microbial community composition related to root type, root section and belowground carbon allocation (study II).** Variation in community composition was assessed after [13]$CO_2$ labeling in the DNA of the [13]C-heavy fraction, representing predominantly [13]C consumers. Variation is visualized by non-metric multidimensional scaling (NMDS) plots based on Bray-Curtis dissimilarities, highlighting the dependence on root type and **A** categorical photosynthate levels defined by [11]C-PET or **B** the mass fraction of [13]C measured in the rhizosphere soil by EA-IRMS. Information on root sections can be inferred from the plots in (**A**), with root tip samples being represented by closed symbols, whereas root base samples are represented by open symbols. Source data are provided as a Source Data file.

example, the bacterial genera *Mizugakiibacter, Chujaibacter, Pseudolabrys, Hyphomicrobium, Porphyrobacter, Catenulispora* and *Massilia* as well as the fungal genus *Mortierella* increased in relative abundance towards the younger roots, whereas a few taxa like *Methylotenera* and *Ralstonia* showed the opposite pattern.

Clustering of samples according to root sections was also clearly noticeable in the NMDS plots, whereas sample clustering according to [11]C-photosynthate allocation was rather weak, especially in study I (PERMANOVA $R^2$ between 0.07 and 0.09). It increased substantially in study II ($R^2$ between 0.34 and 0.47), where we focused on the analysis of DNA in the [13]C-heavy fraction, which represents predominantly the consumers of rhizodeposits. This was reflected in the NMDS plots (Fig. 4A), where samples clustered according to [11]C tracer level as well as root section along the first axis and root type along the second axis. The explanatory power of recent photosynthate allocation in study II was also higher in the unlabeled control samples in this study, but values were not as high as for the [13]C-heavy fraction (mean $R^2$ of 0.28; Supplementary Table 3). For reasons of direct comparability with study I, we applied the same PERMANOVA model in study II (Table 1, Supplementary Table 3), though the explanatory power of the model "[11]C level * root type" was slightly better with even higher $R^2$ values for the factor [11]C level with $R^2$ values of 0.39 (fungi), 0.43 (bacteria) and 0.59 (Cercozoa).

Next, we evaluated whether $^{13}C$ quantified in the rhizosphere was an even better explanatory factor for variation in microbial community composition than the $^{11}C$ tracer signal. The EA-IRMS-derived data on mass fraction of $^{13}C$ reflects C allocation into the rhizosphere over a longer labeling period than the $^{11}C$ signal in the root. The replacement of $^{11}C$ label categories by $^{13}C$ content in the PERMANOVA model resulted in almost equal $R^2$ values of 0.31 to 0.37 for the $^{13}C$ mass fractional data (Table 1). In the NMDS plots, it became evident that sample clustering according to $^{13}C$ content coincided strongly with the clustering by root type and section, which is most clearly visible for bacteria (Fig. 4B). This demonstrates that recent photosynthate allocation in the root and rhizosphere is indeed strongly related to major spatial patterns in the microbiota according to root section and root type, over short periods of time, i.e., hours, as well as over days.

Seeing that quite some variation existed in microbial beta diversity but remained unexplained, especially in study I, we assessed community assembly by iCAMP to gain insight into the relevance of stochasticity in community assembly within the root system (Supplementary Fig. 8). This revealed a strong dominance of stochastic processes, especially drift (68.5–89% for bacteria and 19.8–86.7% for fungi), whereas deterministic processes were less relevant and dominated by homogenous selection (7.2–16.2% for bacteria and 0.16–19.3% for fungi). The homogenous selection was slightly higher for bacterial communities at root tips compared to bases, which became a bit more pronounced when analyzing the community data of the $^{13}C$-heavy fraction in study II. Likewise, homogenous selection gained relevance in fungal communities at root tips in study II. Additionally, dispersal limitation became relevant as stochastic process in the bacterial community at root tips (7.8%), whereas homogenizing dispersal gained substantial relevance in the fungal community at root bases (42.5%).

Study I included an analysis of alpha diversity and abundance. The Shannon diversity indices of the prokaryotic, fungal and cercozoan communities showed significant differences in response to photosynthate level, root type and root section, whereby the prokaryotes showed the strongest variation and Cercozoa the weakest (Supplementary Fig. 9). A reduced diversity was observed with increasing $^{11}C$ allocation, decreasing age of root types and at the root tips. For the prokaryotes and fungi, this was the consequence of a decrease in richness and evenness in response to the $^{11}C$-photosynthate level and root region, whereas the differences related to root type were primarily driven by a decline in evenness. For Cercozoa, changes were primarily related to decreased evenness, but not richness. Concerning prokaryotic and fungal abundance, a strong increase in target gene copy numbers was seen in all rhizosphere samples compared to bulk soil, but strong patterns related to $^{11}C$ allocation or root section were not evident (Supplementary Fig. 10). Only trends were seen, which suggest that higher bacterial abundances appear to be related to higher carbon availability, which was likewise reflected by variation according to root type and region. Further, a negative correlation between alpha diversity and abundance was seen for bacteria (Pearson correlation $r = -0.23$, $p = 0.029$) and fungi (Kendall correlation $\tau = -0.24$, $p = 0.001$) (Supplementary Fig. 11).

## Heterogeneity in photosynthate allocation impacts microbial consumers of $^{13}C$-labeled photosynthates

To reliably identify specific taxa of microbial $^{13}C$ incorporators, we applied further statistical analysis with the DESeq2 algorithm. We therefore grouped samples in two different ways. In one approach, samples were grouped into four categories based on the quantified mass fractions of $^{13}C$ in the rhizosphere (Category 1: 1.5–6.9%; Category 2: 6.9–12.3%, Category 3: 12.3–17.7%; Category 4: 17.7–23.1%; Supplementary Fig. 12). This enabled the identification of taxa that became uniformly and thus significantly labeled related to the amount of carbon transferred into the rhizosphere. Alternatively, samples were

grouped according to their origin in the root system, i.e., from tips or bases of seed-borne (primary and seminal roots) and shoot-borne (crown) roots, respectively. This grouping allowed the identification of labeled taxa with consistent label incorporation according to root architecture. Overall, the number of identified taxa ranged from 2 to 18 per amplified clade and grouping category (Supplementary Table 4). More labeled taxa were detected in the clades of bacteria and Cercozoa than fungi. Most taxa identified as $^{13}C$-labeled were identified by both approaches (Fig. 5, Supplementary Fig. 13), underlining that their labeling was strongly related to both, the spatial location of the taxa in the root system and the amount of $^{13}C$ rhizodeposition. Several of the $^{13}C$-labeled taxa had been identified as responsive to $^{11}C$ level and/or root type already in study I, confirming the strong responsiveness of these taxa to one or both of these factors. Among others, the bacterial genera *Paenibacillus*, *Massilia*, *Methylotenera* and *Rhodotorula* as well as the fungal genera *Fusarium*, *Trichoderma* or *Ustilago* were identified in both studies (Fig. 5, Supplementary Fig. 6).

Several fungal amplicon sequencing variants (ASVs) were exceptionally strongly labeled in samples with the highest $^{13}C$ signature in the rhizosphere soil, while their labeling was significantly lower than that of bacteria and Cercozoa in samples with low $^{13}C$ content in the rhizosphere (i.e., label categories 1 and 2; Fig. 5A). The alternative approach, during which $^{13}C$ incorporators were identified upon sample grouping based on their origin in the root system, showed that this intensive fungal labeling occurred at root tips (Fig. 5B), and thus, taken together, at root tips with highest carbon allocation. Bacterial and cercozoan taxa showed less variation in label intensity between the four defined categories, regardless of the underlying sample grouping strategy (Fig. 5A, B).

Visualizing $^{13}C$ consumer label intensity for each significantly labeled taxon revealed that several ASVs represented $^{13}C$ consumers of the bacterial class *Bacilli*, especially the genus *Paenibacillus* was identified in samples with high $^{13}C$ content in rhizosphere soil, i.e., in $^{13}C$-label category 3 and 4 (Fig. 5C). In contrast, almost all $^{13}C$-labeled ASVs within the *Gammaproteobacteria* (e.g., *Pseudomonas*, *Methylotenera*, *Massilia* and unidentified *Oxalobacteraceae*) and *Actinobacteria* were detected in the weakly labeled category 1 samples. The *Bacilli*, especially *Paenibacillus*, showed the highest relative abundance in the labeled fraction of root tip samples (Supplementary Fig. 14), whereas the *Gammaproteobacteria* with *Massilia* as well as unidentified *Oxalobacteraceae* and *Comomonadaceae* were particularly prominent in root base samples (Supplementary Fig. 15). For some taxa, a further differentiation between seed- and shootborne root bases was seen, especially in the case of ASVs from the Rhizobium group, occurring preferentially as labeled at the seed-borne root bases.

Among the fungi, the most intensively labeled ASVs were represented by the class *Sordariomycetes*, especially the genus *Fusarium* (Fig. 5C), resulting in the high fold change for fungi in $^{13}C$-label category 4 (Fig. 5A). This category is represented by tips of seminal and crown roots of the first node (Supplementary Fig. 12), but *Fusarium* occurred also in the $^{13}C$-heavy fraction of other root tips (Supplementary Fig. 14). Also striking was the $^{13}C$ labeling of some ASVs representing non-identified members of the *Lobulomycetes* (*Chytridiomycota*) (Fig. 5C) with high relative abundance in rhizosphere soil samples of $^{13}C$-label category 1–3, i.e., at root bases and tips of the crown roots (Supplementary Fig. 14). Among the Cercozoa, most $^{13}C$-labeled operational taxonomic units (OTUs) represented members of the *Cercomonadida* and *Glissomonadida*, but in contrast to the highlighted bacterial and fungal taxa, these did not show very particular enrichment or abundance patterns related to the mass fraction of $^{13}C$ in the rhizosphere or related to the root system (Fig. 5). Their labeling and relative abundances were more balanced.

Lastly, we combined the amplicon data of the three microbial groups and explored the role of $^{13}C$-labeled taxa within the community by network analysis (Fig. 6). This was done separately for the root tip

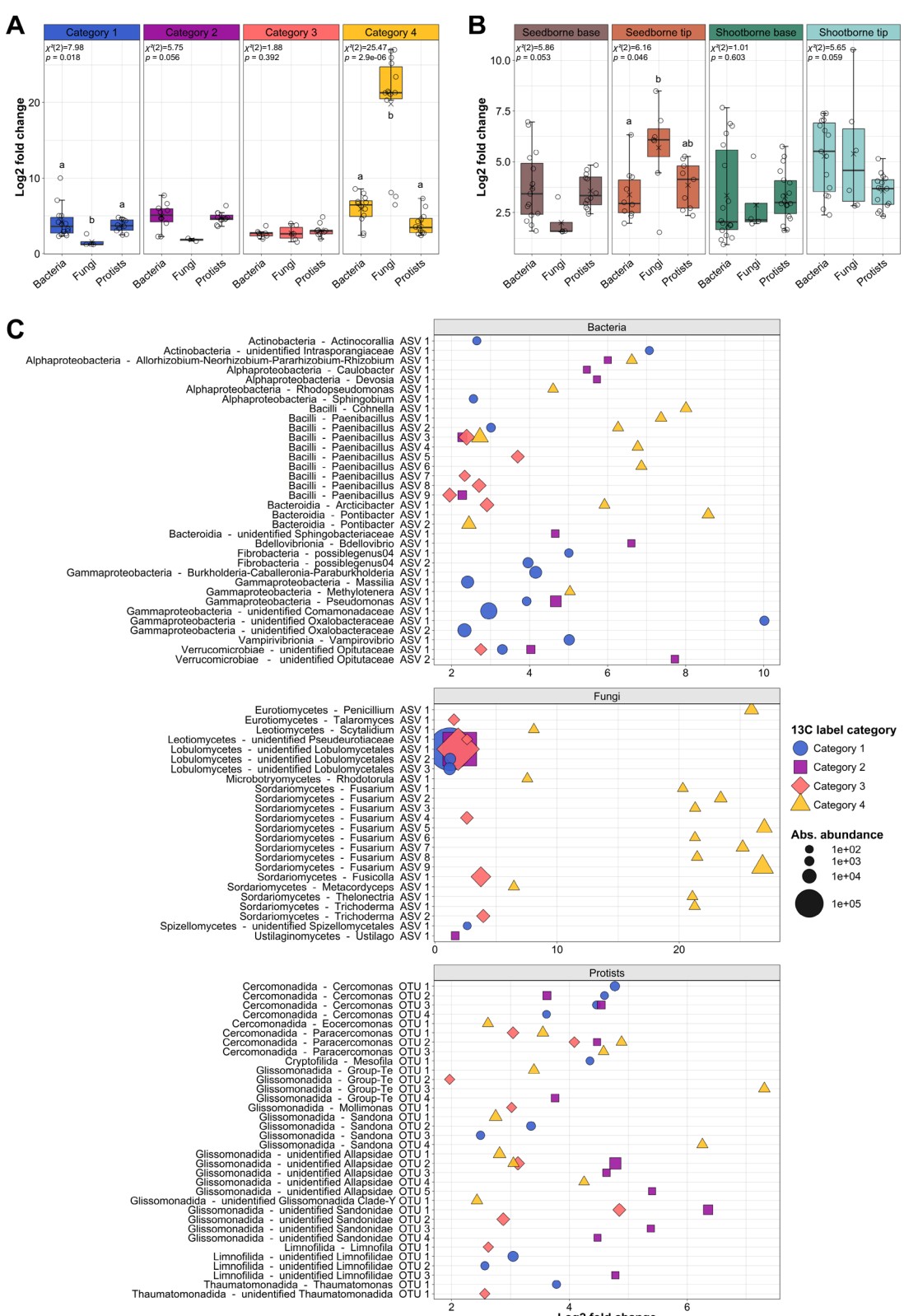

and root base microbiota, as major differences were observed between samples from these two root sections in the SIP amplicon dataset (Fig. 4). Networks were calculated at species level resolution and aggregated at phylum (bacteria, fungi)/order (Cercozoa) level (Fig. 6) or at genus level (Supplementary Fig. 15). Compared to the network constructed for the root tip (49 associations), the network for the root base showed more associations between bacteria and Cercozoa (63 associations), especially between bacteria and *Glissomonadida* (22 and

29 associations, respectively) (Fig. 6). In contrast, many fungal taxa disappeared from the root base network. To assess the role of the $^{13}C$-labeled taxa within these networks, we analyzed their connectivity and compared it to non-labeled taxa. While there were no significant differences observed in the root tip network, the labeled taxa showed a higher degree, radiality and centrality in the network of the microbiota associated to the root bases (Supplementary Table 5), indicating that these had more prominent positions in the network.

**Fig. 5 | ¹³C label intensity and identity of microbial taxa with significant label incorporation.** Significantly labeled taxa were identified using the DESeq2 algorithm of the HTSSIP package. For this analysis, samples were grouped into four categories, either **A** and **C** according to the mass fraction of ¹³C quantified in the rhizosphere soil (Category 1: 1.5– 6.9%; Category 2: 6.9–12.3%, Category 3: 12.3–17.7%; Category 4: 17.7–23.1%; Supplementary Fig. 12) or **B** based on the sample origin within the root system (seed-borne root tips, seed-borne root bases, shoot-borne root tips, shoot-borne root bases). **A**, **B** show the label intensity of significantly labeled taxa according to the fold change in sequence read abundance between the ¹³C-heavy and ¹²C-heavy fraction. Boxes span from the first to the third quartiles, the line inside each box represents the median; x represents the mean and the whiskers extend to the last data point within 1.5 times the inter quartile range. Data points outside of whiskers represent outliers. Significance of differences within subsets was tested by Kruskal-Wallis rank sum test. Distinct lowercase letters indicate significant differences between groups as tested by Dunn's post hoc tests with Benjamini-Hochberg correction ($p < 0.05$). From left to right, **A** consists of $n = 14, 4, 10, 10, 2, 11, 7, 6, 9, 12, 14, 12$ biological replicates and **B** of $n = 15, 4, 12, 10, 7, 9, 18, 4, 18, 15, 6, 15$ biological replicates. **C** illustrates the identity of the labeled taxa along with the enrichment in the ¹³C-heavy fraction over the ¹²C-heavy fraction based on log2 fold changes and the summed abundance of the ¹³C-labeled taxa in the ¹³C-heavy fraction based on read counts. Absolute abundance equals the sum of all reads of a labeled ASV over all samples of a label category. Bacteria and fungi are labeled as Class_Genus_ASV and protists as Order_Genus_OTU. Source data are provided as a Source Data file.

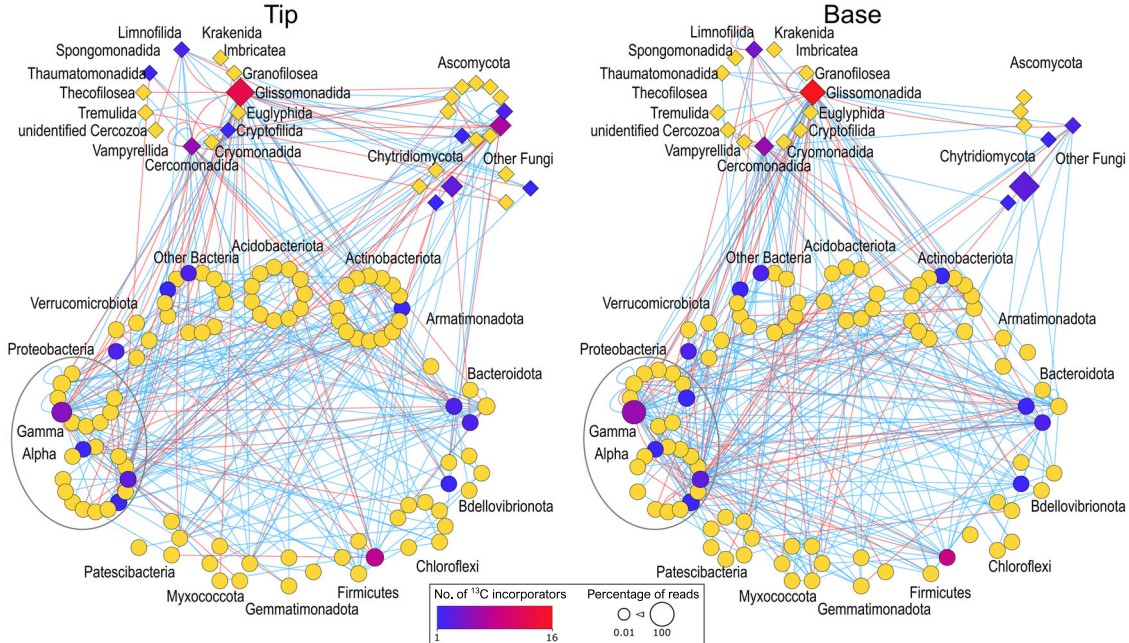

**Fig. 6 | Core networks illustrating the associations between bacterial, fungal and cercozoan taxa and the positioning of ¹³C labeled taxa.** Networks were independently calculated for the root tip and root base microbiota based on ¹³C amplicon sequence data of study II. Nodes are arranged at phylum (bacteria, fungi) and order level (Cercozoa). Positive associations are indicated by blue lines, negative associations by red lines. Network taxa are colored according to the number of ¹³C incorporators (blue to red) or non-incorporators (yellow) as determined by DESeq2 analysis. The node size is proportional to the percentage of reads. Source data are provided as a Source Data file.

## Discussion

¹¹CO₂ labeling in combination with PET and MRI allowed the integration of tomographic data on photosynthate distribution with root structural information at high spatial and temporal resolution[21]. Applied to 6-day-old maize plants, it revealed that recently assimilated photosynthates remained in the shoot and were not translocated into the roots, which is explained by the early developmental stage at which the root system is still supplied by seed reserves[25]. In 13- and 20-day-old plants, the recent photosynthates were rapidly transferred into the entire root system. The primary root received the recently fixed carbon most rapidly after the ¹¹CO₂ pulse, indicating an efficient reorganization from seed- to shoot-derived carbon supply until day 13. It is assumed that the reorganization from heterotrophy to autotrophy occurs in maize around day 10[25]. The observed fast transition ensures the further development and functionality of the seed-borne roots. At the two later time-points, we observed the hypothesized heterogeneities in carbon allocation within the root system.

Within the individual roots, the allocation of recently fixed carbon varied strongly along the longitudinal root axis with most intensive accumulations at the root tips. This was seen in ¹¹C-PET images (Fig. 2A) and confirmed by ¹³C analysis of the root tips as well as for the associated rhizosphere soil (Fig. 3). The intensive photosynthate

accumulations in and around root tips align with previous studies using ¹⁴C[26,27] and ¹³C approaches[15] and can be linked to the energy demanding processes occurring at root tips, including cell division in the apical meristem, mucilage synthesis at the root cap and membrane transport. The related high photosynthate accumulations in the rhizosphere are the consequence of different rhizodeposition processes with photosynthate loss at the root tips due to exudation, mucilage production and border cell shedding[9,26,28,29].

We also detected clear differences in photosynthate allocation between root types, whereby the youngest crown roots showed particularly high ¹¹C signal intensity, especially at their tips. This indicates that these young roots are already well connected to leaves via the phloem and efficiently supplied by recent photosynthates, supporting their high elongation rates[30] and dominant role in soil exploration[31]. The ¹³CO₂ labeling confirmed that photosynthate accumulations at root tips decreased consistently for root types of increasing age (Fig. 3B). This can be explained by the development of lateral roots as additional local sinks, therewith reducing the initially very high accumulations at the main root tip. It is assumed that both, root elongation and the emergence of lateral roots contribute to the total photosynthate sink strength of a root, and these processes are thought to be regulators of root photosynthate transport in the root system[32,33].

At the root bases, the recent photosynthates merely passed through according to [11]C-PET imaging (Vid. S1). Also, the [13]CO_2 labeling over several photoperiods showed that root bases did not accumulate much of the [13]C tracer, especially not the primary and seminal roots (Fig. 3). These roots were already established with the full length that we sampled for the base section when the [13]CO_2 labeling period began. In the younger roots, part of the sampled base section developed during the [13]C labeling period and incorporated [13]C-labeled photosynthates into the tissue, explaining the consistent increase in [13]C content of root samples with decreasing age. Further, the mass fraction of [13]C within the root system was very closely correlated to that in the rhizosphere at the root bases (Fig. 3D) pointing to tightly coupled processes for photosynthate allocation within the root and their release into the rhizosphere. This aligns with our hypothesis and the knowledge about reduced rhizodeposition with increasing root maturity along the root axis, which comes with a stricter control of diffusion and increasing relevance of specific secretion mechanisms[9,34]. These obviously link root internal photosynthate allocation tightly with allocation in the rhizosphere. At the root tips, the root internal and external [13]C tracer accumulations were not well correlated (Fig. 3D), merely because the rhizosphere samples from different root types showed less distinct patterns and more variation between replicate samples than the root tissue samples did. Differences in microbial carbon mineralization might have contributed to this discrepancy, considering that we observed root type specific variation in the microbiota at the root tips (Fig. 4, Supplementary Fig. 5).

As an asset of the temporal resolution of [11]C-PET imaging, we observed heterogeneity regarding the arrival of the short-lived [11]C tracer signal in the individual roots. This may indicate differences in flow velocity between individual roots. In combination with the variation in photosynthate accumulation at root tips, it indicates that specific mechanisms play a role in determining the distribution of recent photosynthates within the root system. Intensely debated in this context are functional differences between the roots[15], root length and diameter[15,28], as well as root growth rates[35]. Further, anatomical traits such as phloem diameter and processes including phloem loading and unloading may contribute to this.

As rhizodeposits present a major energy source for microorganisms[36] and microbial taxa have specific substrate preferences in the rhizosphere[2], we hypothesized that the observed variation in photosynthate allocation within the roots and into the rhizosphere should have consequences for the rhizosphere microbiota. We have multiple lines of evidence that the spatial patterns observed in the microbiota according to root type and root section were closely related to photosynthate allocation. Root section, root type and the two isotopic tracer levels, taken as a proxy for recent photosynthate allocation in the root system, explained jointly variation in microbial community composition according to PERMANOVA. Likewise, the NMDS plots revealed that sample clustering occurred according to root section and root type, and in addition to the [11]C tracer level and the amount of [13]C quantified in the rhizosphere. The taxa that were identified as significantly labeled by the DESeq2 algorithm were largely the same, independent of the grouping of samples according to their origin in the root system or the [13]C tracer category.

The most striking differences in photosynthate allocation were observed along the root axis with accumulations at the root tips (Fig. 2A). These patterns were reflected by differences in community composition of bacteria, fungi and Cercozoa between root tips and bases (Fig. 4, Supplementary Fig. 5) and a reduced alpha diversity of prokaryotes and fungi at root tips (Supplementary Fig. 9). Differences in the microbiota between root tip and base reflect stages of a successive community developmental process along the axis, related to rhizodeposition processes[10,37]. The population size of bacteria and fungi in the rhizosphere did not differ substantially between root base with low recent photosynthate accumulation and root tips with high

accumulations, indicating a very rapid population build-up at root tips, supported by the high substrate availability at root tips[38–40]. This comes along with a decline in alpha diversity, i.e., reduced richness and evenness, explained by a specific enrichment of selected fast-growing taxa[41], such as *Paenibacillus* or *Fusarium*, as identified by DNA-SIP.

Root type was the second major factor that differed in recent photosynthate allocation and to which variation in microbial community structure was related (Table 1). Evidence for root type dependent differences in microbial communities of cereals is recently accumulating[11,42,43]. For maize, differences in bacterial communities have been reported to exist between primary and crown roots[12]. Beyond these root type related differences, we demonstrate with our results a successive change in community composition from the oldest, primary root to the most recently emerged crown roots for bacteria, Cercozoa and, with more variability, for fungi (Fig. 4, Supplementary Fig. 5). Similarly, a successive decrease in alpha diversity from the oldest to youngest roots was seen, especially for bacteria, primarily a decline in evenness (Supplementary Fig. 9). These successive changes can also be explained by community development, based on the observation that several taxa showed consistent increases in relative abundance across root types and related to the chronology of root emergence (Supplementary Fig. 7, Supplementary Data 1, Supplementary Table 2). The development is likely driven by different processes, including root type specific exudation patterns, which have been reported to exist in maize[12], and time-related processes as roots and rhizosphere mature. The sequential emergence of the roots gives more time for community assembly, competitive outcomes, plant-selection processes and top-down control by predators along the root axis towards the bases for older roots compared to younger roots. Our samples from root bases include this temporal aspect, as they differ not only with regard to root type, but also in rhizosphere age.

Root type related patterns were not only observed at the root bases but likewise seen at the root tips. While this can be seen for bacteria in study I, the results of study II show this even more clearly for the communities of bacteria, fungi and Cercozoa that profited from rhizodeposits (Fig. 4A). This demonstrates that root-type related differences in the microbiota develop already at a very early stage of rhizosphere establishment. Considering that differences existed between all analyzed root types including the different generations of crown roots, the underlying mechanism leading to these differences is not necessarily only related to root type, but likewise defined by further processes, probably related to root length, root age or root growth rate[44].

Despite reproducible spatial patterns in the rhizosphere microbiota, substantial variation in beta diversity remained unrelated to root type, root region and carbon allocation patterns. The quantitative analysis of community assembly processes revealed a strong dominance of stochastic processes, especially drift (Supplementary Fig. 8). This may appear unexpected, considering the important role of rhizodeposits in shaping the rhizosphere microbiome, but high variability in the rhizosphere microbiota between individual plants and substantial drift, in part along with other stochastic processes, was likewise seen in other pot experiments and enforced under certain growth conditions and during early plant growth stages[45–47]. It indicates that stochastic processes such as drift are likely to be high in small-scale (pot) experiments and to increase further at even smaller spatial scales, i.e., within a root system of an individual plant. It can result from processes such as competitive exclusion and priority effects, which have been proposed to contribute to microbiome assembly especially at root tips due to the rich amount of rhizodeposits that are released[41]. However, our data showed that stochastic processes and more specifically drift is nearly equally relevant at root tips and bases. Instead, we observed a slight increase in homogenous selection at root tips, which was enforced in the [13]C-heavy fraction of study II. It also increased with inclining [13]C label intensity in the rhizosphere (Supplementary Fig. 8B). This indicates that part of the microbial community becomes

specifically enriched by deterministic processes related to rhizode-position, whereas the enrichment of many other taxa is affected by stochastic processes, therewith introducing heterogeneities. The [13]C-heavy fraction of the bacterial community showed in addition dispersal limitation at root tips, which may also be related to priority effects, when early-arriving taxa limit the establishment of later-arriving taxa. In contrast to the bacteria, the [13]C-labeled fungal communities showed high homogenizing dispersal at root bases under conditions of more limited carbon supply. Fungi may compensate spatial heterogeneities and carbon limitation in the rhizosphere by their hyphal growth, but not yet at root tips, rather over time at root bases.

The focus on taxa with significant [13]C label incorporation allowed to confirm the fourth hypothesis and revealed an exceptionally high labeling of fungi in rhizosphere regions that received highest levels of [13]C-labeled photosynthates, i.e., the root tips (Fig. 5, Supplementary Fig. 13). While bacteria have traditionally been considered the prime consumers of rhizodeposits, recent studies reported that fast-growing "sugar" fungi can also profit from this plant-derived carbon[48,49]. Our data indicate that this occurs in particular in rhizosphere regions with highest rhizodeposition. Towards the root base, competition between fungi and bacteria for the more limited rhizodeposits increases, giving an advantage to bacteria over fungi, demonstrated by higher log2 fold changes in this root region for bacteria than fungi (Fig. 5A) and the disappearance of fungal taxa from the co-occurrence network. A particularly high [13]C labeling in samples with highest [13]C-photosynthate rhizodeposition was linked to *Fusarium* (Fig. 5C), which was apparently very efficient at obtaining organic carbon from rhizodeposition in these regions, i.e., root tips. The genus *Fusarium* includes different species of commensals as well as potential beneficials[50,51] and maize pathogens[52,53]. It is tempting to speculate that these fungi can establish in the rhizosphere preferentially in regions that provide ample amounts of organic carbon and that do not yet host a very competitive microbiome, conditions they encounter at root tips. Besides, root tips are known to represent a good entry point for pathogens[54]. Along with *Fusarium*, potentially beneficial fungi such as *Trichoderma* were also labeled in samples with high [13]C-photosynthate deposition[55]. In contrast, the very high relative abundance of unidentified *Lobulomycetales* within the labeled fungal community indicates that not all fungi follow the same pattern, because this not yet well-known taxon was prominently present and labeled at all root bases and the tips of shoot-borne roots (Supplementary Fig. 14). The different colonization pattern of this taxon was reflected in the co-occurrence network, where it remained visible in the network of the root base with only positive associations to other taxa, whereas many other fungal taxa disappeared (Figure S15). Members of the *Chytridiomycota*, to which the *Lobulomycetales* belong, have been reported to profit from rhizode-posits in grassland soil and is was speculated that these might be secondary consumers rather than primary consumers of rhizodeposits[49]. Our data for the unidentified *Lobulomycetales* support this assumption considering their rather weak labeling and their position and positive connectedness in the co-occurrence network, suggesting that these fungi may be more competitive as secondary consumers than as primary consumers at root bases.

Among the bacteria, the most intensive [13]C label incorporation was observed for *Paenibacillus* in rhizosphere samples with high [13]C content, which reflects the pattern of *Fusarium* and *Trichoderma* and is well in line with their copiotrophic lifestyle, enabling a rapid pro-liferation in root regions with high photosynthate supply[56]. Indeed, *Paenibacillus* was prominent at both, seed-borne and shoot-borne root tips (Supplementary Fig. 13) and showed high ASV diversity, to which operon heterogeneity within strains likely contributed to some extent[57,58]. The [13]C-labeled taxon with the highest relative abundance among the labeled taxa in the [13]C-heavy DNA fractions was *Massilia*, which became labeled in particular at the bases of shootborne roots

(Supplementary Fig. 13). Further taxa with this pattern included different rhizobia, some of the *Paenibacillus* ASVs and other *Oxalo-bacteraceae*. All these genera have previously been identified as carbon consumers in the rhizosphere by DNA-SIP studies[7,59]. These genera are known to include beneficial taxa[4,60], which may be particularly sup-ported in root regions more distant from the tip, where overall fewer photosynthates are released, but the secretion or diffusion of specific compounds gains relevance compared to root tips[61,62]. These com-pounds may more specifically support beneficial taxa, as reported for *Oxalobacteraceae*, which benefit from flavonoids and can improve plant nutrient aquisition[4]. The fact that labeled taxa had prominent positions within the network at the root bases underlines that these became particularly well established at the root bases upon commu-nity succession along the root axis.

Among the bacteria, not only primary consumers became [13]C-labeled, but also the predatory bacterial genera *Vampirovibrio* and *Bdellovibrio* as secondary consumers. They were labeled at the root bases, where they were detected at rather low relative abundance (Supplementary Fig. 13), but with an intermediate [13]C label enrichment compared to the other [13]C-labeled bacterial taxa (Fig. 5). This aligns with a report that these bacterial predators can be highly active[63]. They may thus represent relevant players in the microbial food webs of the rhizosphere besides eukaryotic predators. The latter, studied here with a focus on cercozoan predators, became significantly [13]C-labeled in all different root regions. Although Cercozoa integrate the [13]C label at a higher trophic level, their mean label log2 fold changes followed clo-sely the patterns of the bacterial communities (Fig. 5A, B). This was also seen in the community compositional variation of the total [13]C-heavy DNA fraction, which reflected very well the pattern of their bacterial prey (Fig. 4A). Further, a tight association of protist grazers with potential bacterial prey was seen in the co-occurrence network (Fig. 6), where the main [13]C-lableled cercozoan consumers of labeled prokar-yotic prey were small flagellates in the *Glissomonadida* and *Cercomo-nadida*. These have short generation times and specific grazing impacts on bacterial communities[64,65]. With their establishment along the root axis, they have likely modulated the labeling patterns of some taxa that serve as prey. These findings indicate significant top-down control of the maize microbiome by predatory bacteria, protists and likely also fungi, especially towards the root bases.

In summary, the combination of isotope-based approaches to document recent photosynthate allocation within the root system and into the rhizosphere along with image-guided destructive spatially resolved rhizosphere sampling allowed to link this allocation with heterogeneities in microbial community structure. The spatial pat-terning in photosynthate allocation within the maize root system is dominated by strong photosynthate allocation at root tips and a tight correlation between root internal and external allocation especially at the root bases. The rhizosphere microbiota responds to this with notable changes in community structure along the root axis and dif-ferences between root types. Thus, photosynthate availability is an important factor driving habitat differentiation within the maize root system and causing spatial variation in the rhizosphere microbiome. Fast-growing taxa, here in particular *Paenibacillus*, benefit strongly from photosynthates at the root tips, likewise as some fungal taxa, including potentially pathogenic fungi like *Fusarium* that exploit these root sections efficiently. Towards the root bases, the microbiota undergoes a succession, whereby other taxa are particularly supported by rhizodeposits, including bacterial taxa with potential benefits for the plant, whereas fungal taxa become less competitive under the more limited photosynthate availability. Further, the development of the microbiota is modulated by a range of prokaryotic and eukaryotic secondary consumers that thrive in the rhizosphere, known or pre-dicted predators that exert top-down control on the community. Community compositional differences exist between root types and between each generation of crown roots, evident at root bases as well

as tips. This points to mechanisms and processes that are not only root-type specific, but change related to the age of the root or the rhizosphere or are related to other traits such as root length. It requires further analyses to resolve the underlying mechanisms in more detail, whereby processes that define the composition of the rhizodeposits, which fuel the microbiota, will require particular attention. Taken together, the existence of spatiotemporal patterns in photosynthate allocation and the composition of rhizodeposits and the resulting differences in rhizosphere microbial food webs implies that processes in the rhizosphere are spatially and temporally defined. It requires spatiotemporal resolution to understand microbiota assembly and precisely assess microbially driven processes within the root system, including those that provide potential benefits for the plant, which may be very locally supported in the root system by the plant. This knowledge is crucial for developing effective management strategies for root microbiomes.

## Methods

### Soil preparation and plant cultivation

Soil columns were established according to Vetterlein et al.[66]. Briefly, 16.7% loam soil derived from a Haplic Phaeozem (Schladebach, Germany) was mixed with 83.3% quartz sand (WF33, Quarzwerke GmbH, Frechen, Germany). This growth substrate was dried, sieved to <1 mm and fertilized[66] before homogeneously filling PVC columns (20 cm height, 8 cm diameter) up to a bulk density of 1.47 g cm$^{-3}$. *Zea mays* seeds (line B73) were surface sterilized in 10% $H_2O_2$ for 10 min, primed in a saturated $CaSO_4$ solution for 3 h and sown at 1.5 cm depth. Plants were grown in a climate chamber for 22 days as described[66], while upholding a volumetric soil water content of 18%.

### $^{13}CO_2$ labeling of plants

Stable isotope labeling to trace photosynthates into the rhizosphere and its microbiota was conducted in two Perspex® chambers adapted from Hünninghaus et al.[7] (Fig. 1). Eight plants were transferred into each labeling chamber at day 15 after sowing and either exposed to $^{13}CO_2$ or unlabeled $CO_2$ (referred to as $^{12}CO_2$). Plants were continuously labeled during the whole 12-h light period for 6 days. Both chambers were initially scrubbed of ambient $CO_2$ by passing air through a soda lime cartridge at each labeling day before $^{12}CO_2$ or 99% $^{13}CO_2$ (Linde GmbH, Pullach, Germany) were pumped into the respective chambers and evenly dispersed by ventilators. In both chambers, constant gas concentrations of $407 \pm 20$ ppm were established using an automated system combining $^{12}CO_2$ and $^{13}CO_2$ gas analyzers ($^{12}CO_2$ = Li-820, LI-COR Biosciences – GmbH, Bad Homburg, Germany, $^{13}CO_2$ = S710, Sick AG, Waldkirch, Germany) (Supplementary Fig. 16). Temperature within chambers was regulated to maintain 18 °C at night and 23 °C during the light period, ±2 °C. At the end of each labeling day, the labeling gases within the chambers were scrubbed by passing through a soda lime cartridge and the chambers were opened to allow for inflow of ambient $CO_2$.

### Radioactive $^{11}CO_2$ labeling and PET-MRI scanning

Plants were supplied with $^{11}C$ ($t_{1/2} \approx 20$ min) labeled $CO_2$ and scanned by PET (Fig. 1) to non-invasively visualize the short-term photosynthate distribution in the root system. Due to the short half-life of $^{11}C$, it is produced on site with a dedicated cyclotron. The night before PET scanning, plants were transferred to the *pheno*PET facility[67] and mounted in an environmentally controlled labeling cuvette connected to the gas exchange system with the roots in the field of view of the *pheno*PET. The shoot was subjected to labeling with ~200 MBq (study I) or 100MBq (study II) $^{11}CO_2$ for 6 min after the start of the 2.5 h PET measurement. Data processing and image reconstruction is based on Hinz et al. 2024[67]. The image reconstruction was done in 5-min frames. Scatter and attenuation corrections were not applied. Images were decay-corrected. PET measurements were complemented by

transferring the plant to MRI to non-invasively monitor root system architecture. We used a 4.7-T vertical bore magnet (Magnex Scientific, Oxford, United Kingdom) equipped with a 21-cm gradient system up to 400 mT/M (MR Solutions, Guildford, United Kingdom) and a 10-cm RF coil (Varian, Palo Alto, CA) as previously described[68]. To counter imaging artifacts observed in this soil, the imaging parameters were set to: Spin-Echo Multi-Slice sequence, Bandwidth = 400 kHz, 4 averages, Field of view = 96 mm, 0.5 mm resolution, 1 mm slice thickness, echo time TE = 9 ms, repetition time TR = 2.8 s. MRI data were analyzed using the program NMRooting[68]. We restricted root tree analysis to the axial roots, i.e., the primary root, seminal roots and crown roots, thus excluding all lateral roots and quantified root length (Supplementary Fig. 17). For further analysis and to enable image-guided sampling, overlays of the PET and MRI 3D-scans were constructed in the MeVisLab environment (MeVis Medical Solutions AG, Bremen, Germany).

### Image-guided root and rhizosphere soil sampling

Before sampling, the soil was brought to the targeted volumetric water content to ensure comparable extension of the rhizosphere. Soil cores were pushed out of the pots and MRI images were used for orientation and identification of root and rhizosphere samples. We took root tip samples of about 2 cm length and root base samples of about 10 cm length to capture the largest possible variation of both root sections (Supplementary Fig. 1) from every root, while distinguishing between the primary root, seminal roots, and crown roots from the first to the third or fourth underground node. The length of the root base sample was slightly reduced for the third generation of crown roots, whereas the fourth generation base samples were not yet available. PET scans enabled us to roughly categorize root samples according to three $^{11}C$ tracer signal intensity levels. Image evaluation by visual assessment resulted in a distinction between root tips with high signal intensity, root tips with medium signal intensity and basal root sections with low signal intensity. Lateral roots were cut off to avoid mixed root type signals. As the youngest crown roots were very small on the day of sampling, only root tip samples were taken from this root type. Rhizosphere samples were taken by dipping the root pieces in 0.3% sterile NaCl solution. To sediment the rhizosphere soil, the suspension was centrifuged at 5000 x *g* and 4 °C for 30 min. Root tissue samples were further cleaned upon dipping by vortexing the root pieces in 0.3% sterile NaCl for 15 s. The pelleted rhizosphere soil samples and the washed root pieces were stored at −80 °C.

Analyses were done in study I for individual roots from a total of three plants, whereas the same type of root sample, i.e., roots of each root type and section, from two plant replicates was pooled in study II to obtain enough material for all analyses (Supplementary Fig. 1). The previously obtained PET-MRI scans ensured that the photosynthate allocation and development of sampled roots were comparable (Supplementary Fig. 3). Using 16 plants in total, this resulted in four replicate $^{13}C$-labeled sample sets and four corresponding $^{12}C$-control sets covering all root types, root sections and photosynthate allocation categories in study II (Supplementary Table 6).

### Determination of mass fractions of $^{13}C$ in root and rhizosphere samples

A subsample of each rhizosphere and root sample was dried for 4 days at 50 °C and ground to a fine powder by hand. Approximately 70 mg of rhizosphere soil and 0.5 mg of root material was used for $^{13}C/^{12}C$ isotope analysis with an elemental analyzer (Flash EA 1112; Thermo Fisher GmbH, Bremen, Germany) coupled to an isotope ratio mass spectrometer (Delta V Advantage; Thermo Fisher Scientific, Waltham, MA, USA). We calibrated results against the reference materials calcite (IAEA 603; $\delta^{13}C$ = 2.46‰) and corn starch (Schimmelmann Research, Indiana University; $\delta^{13}C$ = −11.01‰). Results are presented as mass fraction of $^{13}C$ in percent (mass of $^{13}C$ relative to the total mass of C in each sample), as calculated based on Teste et al. 2009[69].

## DNA extraction

DNA was extracted from ~500 mg of wet rhizosphere sample using the FastDNA™ SPIN Kit For Soil (MP Biomedicals™, Santa Ana, Canada) following the protocol of Tournier et al. 2015[70] with minor alterations ($2 \times 45$ s cell disruption, 100 μl elution volume). The DNA was quantified using the QuantiFluor dsDNA System on a Quantus™ Fluorometer (Promega, Madison, WI, USA).

## DNA stable isotope probing

DNA-SIP was performed for 36 $^{13}C$-labeled samples and 36 corresponding $^{12}C$-control samples following the protocol of Lueders et al. 2010[71] with minor alterations. Per sample, ~1.5 μg of DNA was loaded onto CsCl (≥99.999% p.a., Carl Roth, Karlsruhe, Germany) suspensions with a starting density of 1.721 g ml$^{-1}$. Samples were randomly assigned to different ultracentrifuge runs to compensate for possible run-related variation. Gradients were spun at 20 °C and $177.000 \times g$ for 38 h in an Optima™ XPN-80 ultracentrifuge with VTi 65.2 rotor (Beckman Coulter, Brea, CA, USA). Gradients were fractionated from bottom (heavy) to top (light) into 12 fractions (~400 μl) by displacement of the gradient solution with bromophenol blue stained DEPC-water. The buoyant density of each fraction was determined by measuring the temperature-corrected refractive index (nDTC) of a small sample aliquot on an AR200 refractometer (Reichert, Depew, NY, USA). Depending on sample origin (root type, root section), we observed that light, unlabeled DNA peaked at densities around 1.705–1.710 g ml$^{-1}$ and heavy, $^{13}C$-labeled DNA peaked at buoyant densities of around 1.720–1.728 g ml$^{-1}$ (Supplementary Fig. 17). The DNA in each fraction was precipitated by adding 2 Vol of 30% PEG 6.000 in 1.6 M NaCl and 1 μl of glycogen (20 μg; Roche, Basel, Switzerland) before incubating the samples at room temperature for 2 h. This was followed by 40 min of centrifugation at 4 °C and $21.000 \times g$, a washing step in 70% EtOH and a second centrifugation step of 25 min before the pellet was eluted in 25 μl of 10 mM Tris-HCl and stored at −80 °C. After centrifugation, one heavy and one light fraction of each sample was selected for amplicon sequencing. The selection was done individually for bacteria, fungi and Cercozoa based on quantitative PCR (qPCR) data generated for all fractions. When DNA was similarly abundant in two neighboring fractions, the more extreme one was selected for sequencing - the lighter fraction with lower density and the heavier fraction with higher density. Plotting the 16S rRNA, ITS1 and 18S rRNA copy numbers against the buoyant density revealed the fractions that contained peaks of $^{13}C$- and $^{12}C$-DNA, respectively (Supplementary Fig. 17).

## Quantitative PCR and amplicon sequencing

qPCR assays were performed as previously described[72] using 10-fold diluted DNA solutions with slightly adapted thermal cycling protocols (Supplementary Table 7) and the same primers as for amplicon sequencing (Supplementary Table 8).

Prokaryotic 16S rRNA gene amplicons, fungal ITS1 amplicons and cercozoan SSU/18S rRNA gene amplicons were generated following group-specific two-step PCR protocols (Supplementary Table 8, Supplementary Table 9, Supplementary Table 10). As ITS1 amplification products showed several unspecific bands on a 1.5% agarose gel, correct bands were excised after the first round of PCR and purified using the NucleoSpin® Gel and PCR Clean-up Kit (Macherey-Nagel, Düren, Germany). Prokaryotic and fungal amplification products were quantified using the QuantiFluor dsDNA System on a Quantus™ Fluorometer and pooled at equimolar concentrations. 18S rRNA gene amplicons were processed using the SequalPrep Normalisation Plate Kit (Invitrogen GmbH, Karlsruhe, Germany). Pooled PCR products were purified with the CleanNA magnetic bead system (GC-Biotech, Waddinxveen, the Netherlands). Library preparation and sequencing of the amplicons was performed by the West German Genome Center (WGGC) and the Cologne Center for Genomics (Cologne, Germany),

on MiSeq instruments (Illumina, San Diego, Canada) generating $2 \times 300$ bp paired-end reads.

## Sequence data analysis

The raw sequence reads were preprocessed using a customized bash script with Cutadapt version 4.2 to demultiplex the samples[73]. Primer trimming was performed using QIIME 2, version 2022.11[74] and denoised ASVs were created for bacteria and fungi using the DADA2 pipeline[75]. The denoising step also comprised forward and reverse read merging and chimera removal. Sequence alignment was performed using the MAFFT software[76]. The SILVA database (version 138) was used for prokaryotic taxonomy assignment[77] and the UNITE database (version 9) for fungal taxonomy assignment[78]. Data were exported into R (version 4.2.3) and analyzed with the packages *phyloseq*[79], *vegan*[80] and *microbiome*[81]. Singletons and reads unclassified above class level were removed from the data sets (Supplementary Table 11).

Cercozoan raw sequence reads were processed using the custom MOTHUR pipeline v.39.5[82]. After demultiplexing and primer- and tag-sequence trimming, the remaining reads were clustered into OTUs using VSEARCH[83] with an abundance-based greedy clustering algorithm (agc) at a similarity threshold of 97%. Clusters with fewer than 214 reads were removed to eliminate potential amplification or sequencing noise[24]. OTUs were assigned to taxa using BLAST+[84] with an e-value cutoff of 1e-50 and the PR2 database[85], retaining only the best hit. Non-target sequences were excluded. Sequences were aligned using the provided template[24] allowing gaps of up to five nucleotides, and cleaned for chimeras using UCHIME[86].

## Statistical evaluation

Alpha diversity was analyzed using rarefied datasets and differences were assessed by ANOVA and Tukey-HSD posthoc tests or Mann-Whitney $U$-tests with Benjamini-Hochberg correction. Beta diversity was assessed by NMDS plots based on Bray-Curtis dissimilarities calculated from rarefied datasets. Differences in community structure were evaluated by PERMANOVA using the function adonis2() of the *vegan* package. Pairwise PERMANOVA was applied to compare microbial community composition between different root types, using the pairwise.adonis() function of the *pairwiseAdonis* package.

To identify microbial consumers of $^{13}C$-labeled recent photosynthates, we employed the HRSIP() function from the R package *HTSSIP*[87]. To increase statistical power, we grouped samples in two different ways. In one approach, samples were grouped into four categories based on their mass fraction of $^{13}C$ in the rhizosphere as measured by EA-IRMS (Supplementary Fig. 12). Alternatively, samples were classified into four categories based on their origin within the root system: seed-borne root tip, seed-borne root base, shoot-borne root tip, and shoot-borne root base. A separate HR-SIP analysis was conducted for each category using non-rarefied sequence datasets. HR-SIP utilizes the differential gene expression analysis from the R package *DESeq2*[88] to identify ASVs that are significantly enriched in high buoyant density fractions (BD of 1.72–1.75 g ml$^{-1}$) of $^{13}C$-labeled samples compared to the high buoyant density fractions of unlabeled control samples[87].

We conducted co-occurrence network analyses to explore associations between bacterial, fungal and cercozoan communities in the rhizosphere of the root tip and base sections for the $^{13}C$-labeled plants. Prior to network calculation, sequence read counts were summarized at species level. Species that were present in less than 1/4 of all samples were summed into one pseudo taxon to reduce spurious edges[89]. Additionally, the location of the sample within the root system and the mass fraction of $^{13}C$ in the rhizosphere were included in the network calculation. To account for the compositionality of community data, the bacterial, fungal and cercozoan datasets were individually normalized by centered log-ratio transformation. Further,

z-transformation was applied to the numeric metadata, while categorical metadata were hot-encoded. The networks were calculated using FlashWeave v.0.18.0[90] with parameters for homogeneous data (sensitive = true and heterogeneous = false) and without normalization. The species networks were summarized in R to genus and order level and visualized in Cytoscape v.3.9.0[91]. The node parameters were assessed with the NetworkAnalyzer[92] implemented in Cytoscape.

Microbial community assembly mechanisms were analyzed using the icamp.big() function from the *iCAMP* package in R, applying phylogenetic bin-based null modeling with bin-specific confidence intervals and βNTI as turnover metric. To minimize phylogenetic bias, null model randomizations were confined within taxonomic bins (max size: 12). Relative contributions of ecological processes were compared across root sections and $^{13}$C label categories, with statistical significance assessed via 1000 bootstrap iterations using the icamp.boot() function.

### Reporting summary
Further information on research design is available in the Nature Portfolio Reporting Summary linked to this article.

## Data availability
The MRI/PET dataset is available at https://doi.org/10.26165/JUELICH-DATA/Q4NWVE. The amplicon sequencing data generated in this study have been deposited in the Sequence Read Archive (SRA; bioproject number PRJNA1145186). Processed sequencing data (as .RDS) and representative sequences for each ASV (as .FASTA) are available at https://doi.org/10.5281/zenodo.16654991. Source data are provided with this paper.

## Code availability
Custom scripts and all required data files to run them are available at https://doi.org/10.5281/zenodo.16654991.

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

## Acknowledgements

We would like to thank Marius Röder (Molecular Biology of the Rhizosphere, University of Bonn) for excellent technical assistance. Esther Breuer, Marco Dautzenberg and Gregor Huber (Enabling Technologies, IBG-2, Forschungszentrum Jülich GmbH) are acknowledged for their valuable assistance during labeling experiments. We furthermore thank Kirsten Unger (Soil Sciences, University of Bonn) for performing the EA-IRMS measurements. Caroline Marcon and Frank Hochholdinger (Crop Functional Genomics, University of Bonn) are acknowledged for providing maize seeds for this study. The Next Generation Sequencing Core Facility of the Medical Faculty at the University of Bonn is acknowledged for providing the instrumentation for MiSeq sequencing. This work was conducted within the framework of the priority program 2089, funded by the Deutsche Forschungsgemeinschaft (DFG, German Research Foundation)—P18 (Project ID 403637614, C.K. and M.W.) and P14 (Project ID 403635931, M.B.). Michelle Watt currently holds the Adrienne Clarke Chair of Botany, which is supported through the University of Melbourne Botany Foundation.

## Author contributions

S.R.S., R.K., M.W. and C.K. conceptualized the study. S.R.S. performed the experiments and data analysis. L.R., D.N., and K.F. contributed to amplicon and qPCR data generation and analysis. M.F.B., K.F., J.F., M.B. and S.L.B. contributed to data analysis and interpretation. A.C. and S.R.S. performed $^{13}CO_2$ labeling. D.P. and D.v.d. performed MRI measurements and contributed to the data analysis. R.M., C.H. and A.C. performed $^{11}$C-PET labeling and contributed to the data analysis. S.R.S., R.K. and C.K. wrote the manuscript with input from all authors.

## Funding

## Competing interests

The authors declare no competing interests.
