## [Transparent Peer Review file · Nature Communications]

Photosynthate distribution determines spatial patterns in the rhizosphere microbiota of the maize root system

Corresponding Author: Professor Claudia Knief

Version 0:

Reviewer comments:

Reviewer #1

(Remarks to the Author)

Summary: Root microbiomes have previously been shown to depend on root age and spatial location on a given root. Concurrently, root exudation is thought to differ among root types, age, and spatial location. The authors combine a novel ^{11}C PET + MRI imaging with ^{13}C SIP approach to resolve patterns in spatiotemporal and developmental allocation of photosynthates and their contributions to rhizodeposition and microbial community composition. In agreement with previous literature, the authors demonstrate that root tips are photosynthate hotspots. They also show that corn seedlings can allocate more photosynthates to younger crown roots through ^{11}C and ^{13}C measurements. Using ^{13}C DNA-SIP, the authors correlate with ^{11}C photosynthate level, root age, and root section. They demonstrate that the bacterial and fungal communities differ for these different categories. Certain taxa are enriched at root tips due to the high flux of photosynthates, and this community composition shifts to other taxa potentially due to changes in exudation transport and chemistry that may favor secondary consumers.

Major comments:

A major advantage of the approach is the direct relationship of ^{11}C allocation over the physical root structure, yet relationships of root size (length and width) to ^{11}C dynamics were not fully evaluated. For example, lines 127-129 suggest an approximately 50% increase in the time it takes for photosynthate to reach root tips in 20 day old versus 13 day old plants. Is this simply correlated with total root length, or somehow a developmentally regulated process? Similarly, some interpretations of the observed differences in root types or heterogeneity among individual roots might be simply explained by variation in root width, which for example increases along successive nodal root whorls. Lines 430-436 and 490-493 of the discussion touch on this sparsely, but I would not agree without further evaluation that the presented data indicate any regulatory mechanism for distributing photosynthate across the root system.

Due to the short half-life of ^{11}C , the total ^{11}C in the experimental system rapidly decays over the measurement. This decay leads to fewer signal decay events at each subsequent 5-minute interval relative to the actual amount of C. Are ^{11}C tracer signal values corrected for this ^{11}C decay from the initial dosage? And can the authors provide reasoning as to why they used qualitative (low, medium, high) versus quantitative analysis of the ^{11}C signal?

Similarly, the authors note that differences in photosynthate transport rate are observed between individual roots possibly due to different regulatory mechanisms. With the 3 levels of photosynthate transport being assigned based on integrals from the 85-90 minute time interval, is it possible that signal has decayed at some roots versus others which may be at their peak? And, could this affect the categorization of the different ^{11}C photosynthate levels? This possible difference in time to signal accumulation peak may be especially problematic given that the authors see photosynthate transport to the older roots like the primary and seminal roots first before the younger crown roots. A plot of the ^{11}C integrals over time at the different ROIs may be helpful for this.

The authors note in the methods that no attenuation corrections were done. I am curious how close the older root tips to the bottom of PET imaging window and whether signal is attenuated in this region and influences the measured ^{11}C signal values.

Can the authors provide justification for using a ~10cm long sample for the “root base”. What fraction of root length for the various roots did this capture? Assuming the roots were different lengths, using a fixed (and seemingly large) value means the root base samples would vary as a fraction of root length and likely also in relative age along the root (i.e. in shorter roots the base sample tissue is probably younger and closer in age to the root tip). If this is the case could such heterogeneity influence interpretation of results?

It would be nice to have a comparison of 11C signal to 13C abundance to show the correlation in addition to the authors comments about similar trends. Is this linear along more root sections suggesting that 11C signal is comparable to 13C abundance measurements?

For the 13C DNA-SIP analysis, one heavy and one light fraction were selected for sequencing. Looking at Figure S16, it seems as though there is a significant amount of DNA in other fractions. In particular, the ITS copy numbers are high at multiple fractions. By picking only one heavy and one light fraction, are there potential biases to the taxa represented due to the excluded fractions?

There are microbes that are slow growing but exhibit high metabolic activities. Is there the potential for the 13C DNA-SIP analysis to be skewed towards fast growing microbes that more readily incorporate the 13C labelled exudates into their DNA?

The authors note that lateral roots were not factored into the analysis. Yet visual inspection of the video and stills show numerous apparently disconnected lateral root tip hotspots that nonetheless were fed photosynthate through an axial root. What fraction of the total PET signal across the intervals are included or excluded from the examined ROI and does it matter? Due to the small pot size for the corn plant, were there signs that lateral roots especially their tips were proximal to the analyzed root sections and contributed to variability?

An additional comment with regards to lateral roots, the authors note that lateral root growth can increase the sink strength of a root system. While the 11C and 13C tracer analysis suggests youngest root types receive more photosynthate, this is only true for the main root after signal equilibration. The older roots may receive more total photosynthate but distribute it towards lateral root growth leading to less signal in the measured root tip and bases.

Minor comments:

Could the authors comment/speculate on why the area immediately proximal to many root tips (elongation zone?) seems to have very low 11C signal relative to more mature regions upstream and the hotspot root tips themselves? Almost like an exclamation point in several cases.

The authors note in lines 425-427 that 13C abundance in root tips from crown 2 and crown 3 are reduced due to the lag period between 13C pulse labeling and sampling. If you were to remove these points from Figure 3 panels C and D, would the fits improve and be significant at that point? I am especially curious how it would improve the fit to the root tip curve in Figure 3D.

In the Figure S1 caption's 6th line, remove exemplary before illustrated.

In Figure 3, the text within the plots is too small and can be hard to read.

In Figure 3, it would be nice to report the slopes of the correlations. Do the slopes differ and would that suggest root tips allocate more to rhizodeposition than root bases?

On page 10 lines 216-218, the authors write “It revealed a couple of abundant genera with consistently increasing relative abundance in the rhizosphere according to sequential root emergence . . .” It is hard to understand the directionality of the correlation from the phrase sequential root emergence. It might be better to be more explicit and say that “increasing relative abundance in the rhizosphere in younger roots.”

On page 11 lines 256-257, the authors note that strong patterns in dependence of target gene copy number to 11C allocation or root section were not evident as shown in Figure S8. While there does not appear to be a statistically significant difference, there seems to be a weak positive trend which would be consistent with the intuition that more rhizodeposition means more microbial activity.

Reviewer #2

(Remarks to the Author)

Please see comments in attached PDF.

Reviewer #3

(Remarks to the Author)

I co-reviewed this manuscript with one of the reviewers who provided the listed reports. This is part of the Nature

Communications initiative to facilitate training in peer review and to provide appropriate recognition for Early Career Researchers who co-review manuscripts.

Version 1:

Reviewer comments:

Reviewer #1

(Remarks to the Author)

The authors do a good job addressing reviewer concerns regarding ¹¹C PET imaging, ¹³C SIP, and microbiome analysis. The study combines PET/MRI imaging with ¹³C labeling and microbiome community analysis and identifies spatiotemporal patterns of carbon allocation, root exudation, and microbial community assembly. These findings are likely to be of interest to the broad readership of Nature Communications. A few minor comments regarding clarity and typos are listed below.

Lines 70-73, "It is unclear....", this sentence is a bit long and confusing. It would benefit from breaking up the separate thoughts for clarity.

Line 94, I think the authors mean "extends" not "extents".

Line 188, I think the authors are referring to seminal roots not lateral roots.

Line 196, I think "monotonic" is the more common form rather than "monotone".

Line 199, delete the period "Mortierella. increased".

Line 559, delete the word "the" from the phrase "It requires the spatiotemporal to understand microbiota assembly"

Reviewer #2

(Remarks to the Author)

Major Comments

The authors have addressed all major criticism. I have requested additional information and made suggestions to improve the clarity and impact of the article, which I hope the authors implement. The work is worthy of publication despite the semi-quantitative nature of the ¹¹C work, primarily due to the use of ¹³C-labelling paired with SIP-DNA. I plan to use some of these material in teaching my soil and rhizosphere microbiology class in Spring 2026, so I believe this will be impactful on the field.

Line Comments

L52 - 56: This is fine, but you might wish to also capture the importance of substrate concentration (i.e., a low dose of a single carbohydrate can elicit a different outcome than a high dose with the same sugar).

L93: Syntax: "Root-internal heterogeneity in C allocation extents..." This hypothesis need to be clarified.

L96: The 4th hypothesis seems to be an extension of the 3rd hypothesis. It could be a separate hypothesis if the authors identify a specific microbial taxon that is selected for by a specific difference in the quality of rhizodeposit. Otherwise, it stands as a relatively obvious extension of the 3rd hypothesis.

L145: "at the root tips was just not significant" <- In the grand scheme of probabilities, the difference between $p = 0.049$ and $p = 0.054$ shouldn't make a major difference in your conclusions. I recommend focusing on the weaker effect (i.e., r value). Note: the symbol for Pearson's correlation coefficient is a lower case "r"

L197-200: Readers will want to know all OTUs that were differentially abundant across your young to old root axis. You should include this in your publication as a Supplementary Table, and also include the representative sequences for each OTU. This will help accelerate the discovery of a consensus among taxa colonizing these different root compartments.

L221- 224: Nice analysis here. Good question and probably some of the best evidence that exists to make such a claim. You might wish to highlight this result in your Abstract. This part of your Abstract is a little fluffy and can be condensed into a single sentence to make room for your report of the conclusion on L221-224:

"Bacterial, fungal and protistan community structure in these rhizosphere soil samples differed depending on root structure and related spatial heterogeneities in carbon allocation. Especially ¹³C-labeled consumers of rhizodeposits, identified by DNA stable isotope probing, were responsive to photosynthate distribution. They showed differences in labelling according to their spatial localization within the root system."

L225-236: Your article is stronger for having completed this analysis. Nice effort.

L237 – 252: My personal preference is to provide the alpha-diversity statistics first b/c they provide a broader view of trends. In your case, I would recommend moving this paragraph upwards, since it dovetails nicely with what we know about alpha-

diversity in the rhizosphere, namely it decreases as one moves from bulk to rhizosphere, because only a subset of taxa are most competitive / adapted to rhizosphere conditions (high C, suppressive plant secondary metabolites etc.).

L266: The phrase 'domain/phylum' is ambiguous here and may confuse readers, as it blends taxonomic rank with methodological grouping. Since you're referring to the different groups targeted by distinct primer sets (e.g., bacteria, fungi, Cercozoa), it would be clearer to use a term like 'amplified clade' or 'targeted taxonomic group'. Alternatively, you could rephrase the sentence to make the methodological distinction more explicit: "Across the different primer sets (bacterial 16S, fungal ITS, and cercozoan 18S), the number of identified taxa ranged from 2 to 18 per enrichment category."

L256 – 283: I recognize that you've put lots of work into further enhancing your manuscript, but I have one more important task: please be sure that all of the ASV IDs and representative sequences (FASTA) are available in your supplementary material. These are key bits of information that will enable subsequent users to map onto the work you have done. It is incredibly helpful and time saving, and will only increase the re-use of the knowledge you have gained.

Figure 5: Use consistent labels, re: "OTU" or "ASV"

L437 – 461: I agree with these new conclusions.

L472 - 478: Sure, but many *Fusarium* are also beneficial to hosts. In general, the pathogens are cheaters who co-opt the trust gained by the beneficials. *Fusarium* is a highly diverse genus that includes not only notorious pathogens (e.g., *F. oxysporum*, *F. graminearum*) but also many non-pathogenic endophytes and even plant growth-promoting strains, including strain level variation (re: <https://www.frontiersin.org/journals/microbiology/articles/10.3389/fmicb.2015.01248/full>). I recommend more nuance here. A subjective focus on pathogens is an age-old mistake when it comes to host-microbe interactions.

L492 – 495: Before concluding this, please verify that your fungal primer set has been validated for capturing AMF. The Glomeromycota are sometimes missed by different ITS sets. The Tedersoo (2017) paper ("PacBio metabarcoding of Fungi and other eukaryotes: errors, biases and perspectives") might be able to assist in this effort.

L514 – 532: Are there ways that Cercozoa may be labeled directly by exudate uptake? Do they ingest solution when they gulp bacteria? I think your conclusion about predation is very interesting and impactful, so if you could strengthen it, then I recommend mentioning it in your abstract. This is the kind of ecological insight that your high-resolution C tracking can provide direct evidence of.

Reviewer #3

(Remarks to the Author)

made.

Reviewer #1 (Remarks to the Author):

Summary: Root microbiomes have previously been shown to depend on root age and spatial location on a given root. Concurrently, root exudation is thought to differ among root types, age, and spatial location. The authors combine a novel ^{11}C PET + MRI imaging with ^{13}C SIP approach to resolve patterns in spatiotemporal and developmental allocation of photosynthates and their contributions to rhizodeposition and microbial community composition. In agreement with previous literature, the authors demonstrate that root tips are photosynthate hotspots. They also show that corn seedlings can allocate more photosynthates to younger crown roots through ^{11}C and ^{13}C measurements. Using ^{13}C DNA-SIP, the authors correlate with ^{11}C photosynthate level, root age, and root section. They demonstrate that the bacterial and fungal communities differ for these different categories. Certain taxa are enriched at root tips due to the high flux of photosynthates, and this community composition shifts to other taxa potentially due to changes in exudation transport and chemistry that may favor secondary consumers.

Thank you very much for the positive and constructive feedback. We appreciate the time you invested and the valuable comments which helped us to improve the manuscript. Please find below our point-by-point responses to all your comments.

Major comments:

A major advantage of the approach is the direct relationship of ^{11}C allocation over the physical root structure, yet relationships of root size (length and width) to ^{11}C dynamics were not fully evaluated. For example, lines 127-129 suggest an approximately 50% increase in the time it takes for photosynthate to reach root tips in 20 day old versus 13 day old plants. Is this simply correlated with total root length, or somehow a developmentally regulated process? Similarly, some interpretations of the observed differences in root types or heterogeneity among individual roots might be simply explained by variation in root width, which for example increases along successive nodal root whorls. Lines 430-436 and 490-493 of the discussion touch on this sparsely, but I would not agree without further evaluation that the presented data indicate any regulatory mechanism for distributing photosynthate across the root system.

Root length and especially width can indeed be expected to contribute to heterogeneities in ^{11}C arrival in individual roots, but do not very well explain the observed major differences in carbon allocation focussed on in this study. To gain better evidence about underlying factors, dedicated analyses will be required. This includes fully quantitative ^{11}C data (see comments about quantification below) and analyses that cover root morphology and anatomy as well as processes such a phloem loading and unloading. We consider this to be beyond the scope of this study but aim to address it in a future study already in the making.

We recognize that we cannot claim the existence of “regulatory mechanisms”. We have deleted “regulatory” (l. 379). Additionally, we shortened our statement on flow velocities as, after a re-visit of our colour scaling, we re-did our ^{11}C images (Fig. 2, Fig. S1C, Fig. S2, Fig. S3, Suppl. Video1). The observed patterns showed a greater variation and would require a more detailed analysis than we can cover in this study (text adjusted in l. 120-126).

Due to the short half-life of ^{11}C , the total ^{11}C in the experimental system rapidly decays over the measurement. This decay leads to fewer signal decay events at each subsequent 5-minute interval relative to the actual amount of C. Are ^{11}C tracer signal values corrected for this ^{11}C decay from the initial dosage? And can the authors provide reasoning as to why they used qualitative (low, medium, high) versus quantitative analysis of the ^{11}C signal?

All analyses and figures underwent a decay correction. This information has been added in the method section (l. 602). We applied a qualitative analysis, because a fully quantitative image analysis

approach had unfortunately not been established at the time the experiments were performed (see comment about attenuation correction below).

Similarly, the authors note that differences in photosynthate transport rate are observed between individual roots possibly due to different regulatory mechanisms. With the 3 levels of photosynthate transport being assigned based on integrals from the 85-90 minute time interval, is it possible that signal has decayed at some roots versus others which may be at their peak? And, could this affect the categorization of the different ^{11}C photosynthate levels? This possible difference in time to signal accumulation peak may be especially problematic given that the authors see photosynthate transport to the older roots like the primary and seminal roots first before the younger crown roots. A plot of the ^{11}C integrals over time at the different ROIs may be helpful for this.

As the data were ^{11}C decay corrected, differences in decay between roots did not affect our results, including sample categorization. Beyond that, decay is only critical when comparing data of different time frames. As we used the data of the same time frame for our categorization, differences in ^{11}C decay remain negligible.

The authors note in the methods that no attenuation corrections were done. I am curious how close the older root tips to the bottom of PET imaging window and whether signal is attenuated in this region and influences the measured ^{11}C signal values.

The axis of the PET detector ring is vertically oriented (in contrast to horizontal orientation in human and most animal PET systems). Consequently, photon attenuation will not lead to differences between top and bottom roots. Instead, we have differences between the center and the pot walls, i.e. ^{11}C signals from roots in the center are underestimated compared to roots at the pot walls. Attenuation correction would require additional measurements by an approach that was not yet routinely implemented at the time the experiments were performed, but still under development. For quantitative assessment, attenuation correction is a prerequisite, but cannot be applied retrospectively to all images generated in this study.

As attenuation correction was still under development when we performed part of our analyses, we based all analyses presented in this study on the same reconstruction pipeline without correction. We are convinced that attenuation has not affected our results, because most roots (except the primary root) were located near the pot wall. In the meantime, we exemplarily validated our ^{11}C sampling categories using attenuation and scatter corrected data (see figure below), but did not encounter any misclassifications due to attenuation effects. Instead, the fact that attenuation correction was not yet fully established has contributed to the decision to group the root samples into only three ^{11}C -label categories.

(A) Simple image reconstruction without attenuation and scatter corrections vs. (B) complete reconstruction using all corrections. Corrections did not affect visual assessment of categories. Tracer accumulations at young crown roots (red boxes) are still obvious and feature a high activity, as was also observed in Fig. 16 of Hinz et al., 2024 (<https://iopscience.iop.org/article/10.1088/1361-6560/ad22a2#pmbad22a2f16>).

Can the authors provide justification for using a ~10cm long sample for the “root base”. What fraction of root length for the various roots did this capture? Assuming the roots were different lengths, using a fixed (and seemingly large) value means the root base samples would vary as a fraction of root length and likely also in relative age along the root (i.e. in shorter roots the base sample tissue is probably younger and closer in age to the root tip). If this is the case could such heterogeneity influence interpretation of results?

In most cases, the root base samples had the full length of 10 cm. The length was only reduced for the 3rd generation of crown roots, whereas the 4th generation base samples were not yet available. We decided to collect root tip and base samples to have the two extreme regions for each root type with the aim to be able to detect maximum possible differences between root types. We would expect the most contrasting samples from the root bases, considering that the latter are from different root types and with differences in rhizosphere age. We discuss these aspects critically (l. 426-427) and added this justification in l.620.

It would be nice to have a comparison of ¹¹C signal to ¹³C abundance to show the correlation in addition to the authors comments about similar trends. Is this linear along more root sections suggesting that ¹¹C signal is comparable to ¹³C abundance measurements?

We agree that this would be an interesting analysis, but as we were not in the position to generate quantitative ¹¹C data at the time the experiments were performed, this analysis cannot be done here.

For the ^{13}C DNA-SIP analysis, one heavy and one light fraction were selected for sequencing. Looking at Figure S16, it seems as though there is a significant amount of DNA in other fractions. In particular, the ITS copy numbers are high at multiple fractions. By picking only one heavy and one light fraction, are there potential biases to the taxa represented due to the excluded fractions?

It happens that DNA peaks in more than one fraction. Looking at all our samples, we see sometimes two adjacent fractions with high DNA loads and sometimes a clear peak in only one fraction, regardless of the target gene. We decided to be consistent in collecting only one light and one heavy fraction for each sample. In cases where DNA was almost equally abundant in two adjacent fractions, we always selected the more extreme samples (i.e. the light fraction was the one with lower density and the heavy fraction the one with higher density). In case corresponding fractions of replicate samples are not well overlapping, we may identify less taxa as significantly labelled. We added this information on the sample selection procedure in I.681-683.

There are microbes that are slow growing but exhibit high metabolic activities. Is there the potential for the ^{13}C DNA-SIP analysis to be skewed towards fast growing microbes that more readily incorporate the ^{13}C labelled exudates into their DNA?

Certainly, differences in carbon use efficiency can affect results, likewise as growth rates do. Fast-growing microorganisms rapidly incorporate ^{13}C -labeled substrate, which can lead to their overrepresentation. On the other hand, slow-growing microorganisms may remain under the detection threshold of SIP. With our relatively long ^{13}C -labelling time of 6 days and a labelling of 12 hours a day we aimed to find a balance between detection of both fast and slow-growing microorganisms, as well as enabling the detection of secondary consumers for the analysis of higher trophic levels (protists). Still, the identified 'labeled' taxa are those that exhibit the highest level of labeling on the day of sampling. Pepe-Ranney et al. (<https://doi.org/10.3389/fmicb.2016.00703>) showed that the phylogenetic composition of carbon metabolizers changes over time, a point that could be addressed by the analysis of multiple sampling timepoints in follow-up studies.

The authors note that lateral roots were not factored into the analysis. Yet visual inspection of the video and stills show numerous apparently disconnected lateral root tip hotspots that nonetheless were fed photosynthate through an axial root. What fraction of the total PET signal across the intervals are included or excluded from the examined ROI and does it matter? Due to the small pot size for the corn plant, were there signs that lateral roots especially their tips were proximal to the analyzed root sections and contributed to variability?

Indeed, the images and videos show ^{11}C -signals for lateral roots, even in cases where these roots were not visible by MRI. Because of the limitation in quantification, we cannot calculate the percentage of ^{11}C tracer in lateral roots from our data. The signals of lateral roots did not affect our classification of samples into the three categories, as this was done semi-quantitatively based on visual assessment of the axial roots only (primary, seminal and crown roots) and obvious presence of lateral root tips close to the categorized root sections would have been noticed. To make sample categorization more accessible, we have complemented figure S1 with an example using plant 2A of study II, featuring 4 crown roots of the 3rd node with high ^{11}C signal intensity (Fig. S1c).

An additional comment with regards to lateral roots, the authors note that lateral root growth can increase the sink strength of a root system. While the ^{11}C and ^{13}C tracer analysis suggests youngest root types receive more photosynthate, this is only true for the main root after signal equilibration. The older roots may receive more total photosynthate but distribute it towards lateral root growth leading to less signal in the measured root tip and bases.

We fully agree with this statement. Please note that we do not claim that lateral roots increase the sink strength, as we cannot prove this quantitatively in this study. We only say that they contribute to the total sink strength of a root (l. 352). We use exactly the explanation given by the reviewer here to explain why older roots with more laterals have less ^{14}C tracer accumulation at their main root tip compared to younger roots.

Minor comments:

Could the authors comment/speculate on why the area immediately proximal to many root tips (elongation zone?) seems to have very low ^{11}C signal relative to more mature regions upstream and the hotspot root tips themselves? Almost like an exclamation point in several cases.

This is an interesting observation that we see in almost all of our PET images. It appears that minor amounts of recently fixed carbon are used for biomass build-up in the elongation zone, but that assimilates are instead mostly transferred to the root tip. The cells primarily stretch in this zone by the uptake of water, accompanied by cell-wall loosening and remodelling (<https://doi.org/10.3389/fpls.2016.01242>), thus the requirement for recently fixed carbon may be lower here. In addition, the strong sink strength of the meristematic zone may cause a local depletion zone of ^{11}C just above the meristem. Even though this is an interesting aspect, underlying reasons remain speculative and we prefer not to add this aspect in the manuscript.

The authors note in lines 425-427 that ^{13}C abundance in root tips from crown 2 and crown 3 are reduced due to the lag period between ^{13}C pulse labeling and sampling. If you were to remove these points from Figure 3 panels C and D, would the fits improve and be significant at that point? I am especially curious how it would improve the fit to the root tip curve in Figure 3D.

We performed the suggested analysis and removed root tip samples from crown root 2 and 3 in the plots above. However, we did not observe a stronger correlation for root tip data when excluding these crown root tips (with recalculated ^{13}C data; see response to reviewer 2). Similarly, the slopes are almost identical to the ones observed in the original figure 3C (slope for full data set: $\beta = 0.1939$, reduced: $\beta = 0.1940$) and figure 3D (slope for root tips full data set: $\beta = 0.1398$, reduced: $\beta = 0.1322$). We explain this as follows: the lag period between ^{13}C pulse labelling and sampling came with a similar delay for all root tips and associated rhizosphere soil samples, regardless of root type. Thus, the effect is largely comparable for all root tips. Instead, the weaker correlation at the root tips results from higher variation of the ^{13}C signal in the rhizosphere samples of older roots, despite low variation within root tips of these roots, evident when comparing Fig. 3A and 3B. We address this in l. 157-158. Besides, we updated Fig. 3C with colors and shapes so that the reader can better distinguish between root types and sections.

In the Figure S1 caption's 6th line, remove exemplary before illustrated.

Modified as suggested.

In Figure 3, the text within the plots is too small and can be hard to read.

Font size enlarged.

In Figure 3, it would be nice to report the slopes of the correlations. Do the slopes differ and would that suggest root tips allocate more to rhizodeposition than root bases?

The slopes have been added to Fig. 3C and D. Differences between slopes in Fig. 3D could indicate that different root types differ in rhizodeposition rates between tip vs base. However, slopes do not differ markedly ($\beta=0.1308$ for root bases and $\beta=0.1398$ for root tips). The fact that root tips contribute more to rhizodeposition than root bases in general, as suggested by the reviewer, can be seen from the higher intercept with the y-axis for the root tip vs base regression line.

On page 10 lines 216-218, the authors write "It revealed a couple of abundant genera with consistently increasing relative abundance in the rhizosphere according to sequential root emergence . . ." It is hard to understand the directionality of the correlation from the phrase sequential root emergence. It might be better to be more explicit and say that "increasing relative abundance in the rhizosphere in younger roots."

We rephrased the preceding sentences and the specific on as follows: "genera with monotone changes in relative abundance in the rhizosphere from older towards younger roots" (l. 195-196).

On page 11 lines 256-257, the authors note that strong patterns in dependence of target gene copy number to ^{11}C allocation or root section were not evident as shown in Figure S8. While there does not appear to be a statistically significant difference, there seems to be a weak positive trend which would be consistent with the intuition that more rhizodeposition means more microbial activity.

High variation in the qPCR data resulted in non-significant differences between mean values. We agree with the reviewer that some trends appear to exist in case of the bacterial data, which suggest higher gene copy numbers in samples with higher ^{11}C tracer signals, at root tips, or for crown roots. Encouraged by the reviewer's comment, we dare to address this in the manuscript as follows: "Only trends were seen, which suggest that higher bacterial abundances appear to be related to higher carbon availability, which was also reflected by variation in dependence on root type and region." (l. 248-250).

Reviewer #2 (Remarks to the Author):

Thank you very much for the positive and constructive feedback. We appreciate the time you invested and the valuable comments which helped us to improve the manuscript. Please find below our point-by-point responses to all your comments.

General comments

The study by Schultes *et al.* provides a high resolution view into the spatial allocation of photosynthate to roots and rhizodeposits, and its structuring of the rhizomicrobiome. Their methodology provided direct measures of the translocation to root hotspots, and their observations are high quality, impactful, and internally consistent. While I found the experiment elegant, I did feel that the hypotheses were fairly generic, so much so that the authors did not bother to explicitly revisit them in their Discussion. As an exploration of hard-to-observe, and poorly described, belowground phenomena, that may be forgivable. The authors should be lauded for studying across domains (scales) of soil fauna and to pair and contrast to two isotope tracing methods.

We rephrased the hypotheses to be more specific regarding our expectations (l. 91-97) and come back to each hypothesis in the discussion (l. 334-335, 367, 384-388, 462-464).

Overall, the analysis and interpretation of results are valid (though I have many clarifying questions in the line comments). However, there is one major exception: the authors refer to '13C abundance' in the text but the measurements they obtained are atom % 13C, which is a proportion of 13C relative to the total C. The conflation of proportion 13C with abundance may impact several of the authors' conclusions and **will need to be addressed**. Accounting for differences in the mass of root (from which the proportion is measured) will be necessary to draw several conclusions made by the authors. While this is a current flaw in their analysis and interpretation, it can be corrected. In fact, I urge the authors to try to provide quantitative estimates, where possible, for the mass of photosynthate accumulating among these different root types. In general, the authors default to generalizations of their measures of intensity. Could they explain why they did not attempt to make estimates of abundance? Admittedly, these measure could be biased by assumptions but given the MRI data and the capacity to calculate root masses, this could (in theory) be done.

The calculation of quantitative estimates for the mass of recently assimilated photosynthates accumulating in individual roots is critical, because only root sections were used for ¹³C analyses. We consider it too critical to extrapolate our data with the aim to obtain full-length root data from ¹³C data. However, this is a valuable suggestion for future studies with ¹¹C data when we will have quantitative data, which can then be integrated with MRI data on root biomass. What we did in this study is a normalization of ¹³C tracer data with the biomass of each individual sample we analyzed, as proposed by the reviewer. This is now integrated into the manuscript. For further details, please see the comment given below in response to your comment on L157.

In hindsight, the work would have been more impactful if paired with shotgun metagenomic sequencing, since pairing hotspots of rhizodeposition with functional activity would have been very revealing. Yet, this is forgivable, since the effort and expense were substantial.

We share this opinion: shotgun metagenomic sequencing, possibly even of heavy SIP fractions, will provide valuable further insight, but increases workload tremendously, as well as the aspects that can be discussed in a paper. We feel that it can stand as independent follow-up study.

Lastly, there were some missing references that could be particularly useful here: [10.1038/s42003-021-01988-4](https://doi.org/10.1038/s42003-021-01988-4), [10.1016/j.soilbio.2019.05.011](https://doi.org/10.1016/j.soilbio.2019.05.011), [10.1038/s41564-018-0129-3](https://doi.org/10.1038/s41564-018-0129-3), and <https://pubmed.ncbi.nlm.nih.gov/36964135/> (and listed in the **Line Comments**).

We have added these references in the introduction, as specified in response to the line comments.

Line comments

L32: Syntax. Do the authors mean: "...in *the* rhizosphere microbiome"?

Changed from microbiota to microbiome (l. 33).

L43: Rhizodeposit indicates that the communities were from mineral particles, is that correct?

We specified by saying "rhizosphere soil samples" (l. 42).

L45: "...profited most from rhizodeposits in specific root regions." <- This sentence is good, but could you be more specific? Maybe: "...profited most from rhizodeposits in specific root regions *where rhizodeposition was greatest*" or some other description.

This is not exactly what we meant; we rephrased "They showed differences in labelling in dependence on their spatial localization within the root system." (l. 44-45).

L46: The term 'niche differentiation' connotes a specific ecological process, which focuses on how competing organisms will adapt ('differentiate') to a new niche to lessen competition. I do not think this is what the authors have in mind here (I could be wrong). If I'm correct, please use a different choice of words.

We modified the terms and state now: "root photosynthate allocation supports distinct habitats in the plant root system" (l. 46).

L65: Please clarify 'this'. The variation in root types and structure? The current meaning suggests 'Complex root systems likely result in the development of root-type specific microbiomes.' I don't completely follow the logic here: "because roots systems are complex, there should be root-type specific microbiomes." It would be stronger if you argued: 'different root types produce different rhizodeposits (ref). And, root systems are comprised of various root types and architectures (ref). Thus, root systems will have a root-type specific microbiomes.'

Thanks for the suggestion. We have rephrased the whole section and integrated the references mentioned above. We report about the spatial variation in rhizodeposition within the root system, which is likewise seen in the microbiota. We argue that this points to the existence of small-scale selection processes, linking root carbon allocation/rhizodeposition with the rhizosphere microbiome. (l. 59-68).

L69-72: While I am interested in the system you are studying, the justification of the system remains "largely unresolved at the small scale" is not a compelling one. What might we better understand about the soil / rhizosphere microbiome or plant-mediated microbial activity in soil by resolving the spatial variation in the rhizosphere microbiome? In other words, one can argue that we haven't resolved fine-scale dynamics in many environmental microbiomes, why this one? I don't think the authors will have a hard time arguing why, but I'd like them to present a strong justification.

We have added a statement to clarify the necessity of small-scale understanding: "Such small-scale dependencies will guide microbial community establishment and processes in the rhizosphere with possible implications for the whole food-web that is fueled by rhizodeposits and possibly for plant

performance, considering that the rhizosphere microbiome includes taxa with potential benefits for the plant.” (l. 73-76)

L86: Syntax: “...and (III), in particular, microbial consumers...”

Obsolete, hypothesis has been rephrased.

L83 – 86: Similar to the justifications provided, the hypotheses are basic. The authors are doing something never been done before, but the justifications (“it’s never been done”) and hypotheses (“we’ll see heterogeneity, we’ll see growth patterns in the rhizosphere that map to where the carbon is flowing, and microbes that feed on rhizodeposits will benefit most”). The author could provide a more compelling reason for why their high-resolution technique is valuable. For starters, it gives us greater certainty over who is responding to fresh rhizodeposits. One could, in theory, chart diurnal and developmental influences of plant on soil / rhizosphere biology. It’s clearly exciting, but the hypotheses leave much to be desired.

We have rephrased and specified the hypotheses (l. 91-97). We have included the relevance of root types and longitudinal root axis for pattern formation. Further, we emphasize that root-internal carbon allocation is indeed reflected in rhizodeposition, which could have been expected, but has not yet been convincingly proven and is a prerequisite for our further hypotheses. The hypothesis describing the reflection of carbon allocation patterns in the rhizosphere microbiota is extended by a fourth, new hypothesis, in which we specify that “microbial taxa are selectively supported by rhizodeposits related to their localization within the root system”. These spatial patterns can be proven with our data, whereas diurnal and developmental influences require temporally resolved studies of the associated rhizosphere microbiota.

L87: The belowground distribution of photosynthate to be spatially and temporally dynamic and that these patterns would produce variation in rhizosphere extent and activity. A review from 5 years ago make a similar case: [10.1016/j.soilbio.2019.05.011](https://doi.org/10.1016/j.soilbio.2019.05.011). Here are some primary sources: [10.1038/s41564-018-0129-3](https://doi.org/10.1038/s41564-018-0129-3), <https://pubmed.ncbi.nlm.nih.gov/36964135/>. All three are pretty key papers for the topic of the present work.

We have implemented findings and references: Zhalnina et al. 2018 as ref. 2 in l. 54 and 68; Kuzyakov & Razavi 2019 as ref. 8 in l. 62, from above King et al 2021 as ref. 13 in l. 64. The study by McLaughlin et al 2023 has a focus on temporal and plant species specific differences; it does not differentiate within the root system or include microbial analyses. We prefer not to include it in addition.

L90 (also, L112, L138 & L607): How did the authors decide on their sampling schedule of 6, 13, and 21 days?

We decided to scan during the first three weeks to capture the transition from seed- to shoot-derived carbon supply. In MRI/PET, a meaningful analysis for maize older than three weeks is challenging, as individual roots are more difficult to segment and effects due to pot size limit become more relevant. Based on this, we decided for approx. weekly analyses at days 6, 13 and 20, with destructive sampling at day 22.

L94 (also missing from section on L618): How where the photosynthate levels determined?

We determined photosynthate levels by visual assessment. We have modified phrasing in the methods section to clarify it (l. 625-627). Further, we added information to the conceptual Fig. S1. In the meantime, we exemplarily validated our categories using attenuation- and scatter corrected images (see response to comment from reviewer 1).

L112: It might be better to call these 'growth stages' or 'developmental stages', since they correspond to the size of the root system. This will help differentiate between the temporally resolved data you present in Figure 2b / the video.

We replaced "time points" by "plant developmental stages" (l. 108).

L113: What is the sensitivity of the ^{11}C PET scan? Your observation is valid in relative terms, but you need to acknowledge your level of detection.

L116: Did the authors try to estimate how much C was present in the 'high,' 'medium', and 'low' photosynthate classes using the ^{13}C data? Do the authors know whether the 'high' signal is saturating the detector? I am not an expert in PET, so the manuscript would be more informative if the authors helped me understand how quantitative the signal is and what the limitations are. Put simply, what is the rang between 0 and 'max' for your detector?

Both comments relate to the quantification of ^{11}C signals by the PET sensor. We like to give a short overview of the technique, which hopefully clarifies the questions:

The $^{14}\text{CO}_2$ is supplied as a tracer, meaning only a tiny fraction of ^{12}C atoms are replaced by ^{11}C . This fraction can be adjusted to fit the measurement range of our sensor. The amount of tracer is measured in activity with the unit MBq. PhenoPET can measure up to 100 MBq in the center of the phenoPET before it reaches detector saturation (Hinz et al. 2024, DOI: 10.1088/1361-6560/ad22a2). We supplied 200 MBq for 6 min to the plant in form of radioactive CO_2 . Only a fraction of this CO_2 is taken up by the plant and from that only a fraction is reaching the root system. But even if the complete 200 MBq would be transferred to the roots, the activity is reduced over time by decay. For ^{11}C , the half-life is 20.4 min. Thus, after 20 min we have only 100 MBq ^{11}C left in total. Thus, saturation effects can be excluded; if relevant, then only during the first few minutes after the pulse, where label begins to reach the root system. We can measure quantitatively over 9 half-lives (PhD thesis Carsten Hinz, <https://doi.org/10.25926/tqj1-fs08>), thus the dynamic range of phenoPET is at least a factor of $2^9=512$ between detector saturation and lower detection limit.

Regarding quantification of the signal: When all corrections are applied, the measured PET signal is directly proportional to the amount of carbon. At the time when we performed the experiments presented in this study, all corrections except attenuation and scatter corrections were implemented in image reconstruction (see also answers to questions of reviewer 1 above). We cannot apply these two corrections retrospectively to all of our images, as they require a measured attenuation map, but we can describe the expected effects: Scatter mainly leads to a blurring of the PET images that can be corrected. Attenuation becomes important when the radiation is absorbed or scattered by matter before reaching the detector. As outlined in response to reviewer 1, carbon in the center of the soil column is stronger affected by attenuation than carbon at the pot wall, but without implications on the sample categorization we applied in this study.

L132: Can the authors quantify how many of the roots did not receive any ^{11}C label relative to the total root mass? The temporal dynamic is interesting, but so is the observation that some roots do not appear to channel freshly fixed ^{11}C .

We agree that this is an interesting phenomenon. We see this in a lot of our measurements and can only speculate about the reasons. Here, in the temporal projections over the whole imaging period, it appears that roots not receiving ^{11}C belong mainly to the lateral roots. Possible reasons could be: (I) roots are temporarily not supplied with photosynthates, (II) supply to these roots is too slow to visualize recently fixed photosynthates, e.g. because roots are supplied by previously stored carbon (III) supply is below the detection level of the phenoPET system.

L157: The authors refer to '13C abundance' in the text but the measurements they obtained are **atom % 13C**, which is a proportion of 13C relative to the total C (i.e., the remainder being 12C). Thus, the terms 'relative abundance', 'proportion', or 'enrichment' are more appropriate. The authors

should also report total ^{13}C or even 'excess ^{12}C equivalent' normalized to the C mass in their plant tissues (see: <https://esajournals.onlinelibrary.wiley.com/doi/full/10.1890/08-1884.1> for calculations). The mass of each root type will impact their conclusions, given the new influx of fresh ^{13}C label will be diluted to differing degrees proportional to the existing ^{12}C mass. The author should try to estimate root masses for each root type based on the root volume (MRI) using assumptions about the mass and quantity of C in root tissues. For root tissues, this could be tractable, while the rhizosphere might get a little squishy. The authors should explore this in order to gain a better understanding of the carbon dynamics. Arguably, there will be error biases that result from estimations, but this should be at least considered and presented along with the atom % ^{13}C measurements.

Thank you for the hint, we have now calculated the mass fraction of ^{13}C as a percentage of the total C mass (w/w) in each sample, using the suggested formulas (2) and (3) from Teste et al. 2009 (<https://doi.org/10.1890/08-1884.1>) with our data on the amount of sample material used for IRMS and the measured C content of each sample. We checked and updated Fig. 3, Fig. 4, Fig. S4 and Fig. S12 as well as Table 1 and Table S3 accordingly. We changed the term " ^{13}C abundance" to "mass fraction of ^{13}C " throughout the manuscript.

Unfortunately, an extrapolation to the whole root system based on MRI data is difficult, even though we do have a protocol to calculate root traits such as root fresh weight from MRI (van Dusschoten et al. 2016, <https://doi.org/10.1104/pp.15.01388>). However, the current experiments were performed within a large research consortium (SPP2089, <https://www.ufz.de/spp-rhizosphere/>), where a specific soil had to be used. Not all soil types work equally well for MRI root images (Pflugfelder et al., 2017, <https://doi.org/10.1186/s13007-017-0252-9>), and sadly, this specific soil proved to be suboptimal for MRI root images. We had to tweak our normal imaging procedures in order to produce images of sufficient quality to enable MRI-PET overlays (see material and methods). But with these images the established root trait analysis cannot be performed to obtain sufficiently robust root fresh weight data.

L168: "For the tissue of the root tips, the ^{13}C abundance increased with decreasing root age, ..." <- Two thoughts: (i) can the authors write more clearly with less jargon? "decreasing root age" = "younger" and (ii) younger roots will have received more label by default, since the new roots will be comprised of recently fixed C (in your case $^{13}\text{CO}_2$). The value of these observations is limited in my opinion, especially without adequate context about what the authors expected to observe related to their hypotheses.

Thought (i): we improved wording as suggested (l. 147-148).

Thought (ii): We report here results that support our second, now rephrased hypothesis. The rephrased hypothesis should now better justify the necessity of these results. We likewise think that the comparison of ^{11}C and ^{13}C tracer signals is relevant to understand the differences in the data. These are the aspects we focus on, here in the results section and in the discussion.

L168: Could this discrepancy between base root (with a relatively consistent mass) and root tips (with a much more variable mass) play a role in the variation you observe. As in, if you normalize to root mass, will the correlation between enrichment and rhizosphere in root tips strengthen?

After recalculating the atom% ^{13}C to proportion of ^{13}C relative to total C mass, we did not find a meaningful change of the correlation between root C and rhizosphere C (Fig. 3, previous R^2 value: 0.76, updated R^2 value: 0.77).

L171: "The decline in..." <- This sentence could be more clearly articulated. Too many conditional statements and too long.

The sentence has been rephrased and divided into two sentences (l. 150-154).

L202: "...differed in dependence..." <- awkward wording. Try: "...differed according to..."

Corrected as suggested (l. 174).

L203: PERMANOVA does not really 'validate' results – it provides evidence. The authors have done a nice job with their statistical analyses, but some of the wording could be refined.

We have rephrased the sentence (l. 184-185).

L211: "...but a certain successiveness was also evident." <- This is too vague. Please describe the phenomenon. Please do not make me imagine what 'a certain successiveness' means.

We changed the paragraph in response to the next comment and deleted this phrase.

L212: This is repetitive. PERMANOVA is not a measure of sequential change. You might wish to try a Procrustes analysis for this (T1 -> T2 vs. T1 -> all others, then T2 -> T3 vs. T2 -> all others etc.). If the succession from T1->T2 is a better fit, for example, you have grounds to describe a form of succession.

We tested the proposed Procrustes analysis for the ¹³C-heavy fraction of the study II dataset, but are not convinced that Procrustes is very helpful in this context. It requires a uniform sample size of the datasets to be compared, but our ITS dataset does not meet this requirement. We would have to remove data, though the number of replicates is already low for this kind of analysis. Alternatively, we performed pairwise comparisons with PERMANOVA (incl. Bonferroni correction) (Fig. S6). While the limited number of replicates is also problematic here, the obtained R-values, though admittedly mostly non-significant, still reflect very well the successive pattern related to root types that we see in the NMDS plots. Seeing the reproducible patterns in the NMDS plots in both studies for bacteria, fungi and cercozoa and the monotone changes in relative abundance of specific taxa, we are confident that we can state that the gradual shift in community composition follows the developmental trajectory of root types, being indicative of a directional, though not strictly stepwise, microbial succession. We have rephrased the paragraph to justify this more convincingly (l. 180-200).

L216-220: These are nice observations, but it would be best to back up your claims about 'sample clustering ...reflecting the chronology of root emergence.'

The described observations were statistically evaluated as specified in the legend of Figure S7. We have now added the detailed results of the posthoc tests in Table S1 and Table S2.

L223 - 228: This is strong evidence communicated effectively. Nice job. It is a nice demonstration that your methods worked for capturing variations in the hotspots of rhizosphere activity.

Thanks for this positive comment.

L285 – 289: The description is not clear. I believe I understand what the authors did, but it took some head scratching. Please clarify.

We rewrote the section to clarify how we analyzed our data (l. 257-265).

L288: The authors have used "root type" and "root section" consistently up until now. What is 'the origin in the root system'? It might be useful to give examples in a bracket, like: "Alternatively, samples were grouped according to their origin in the root system (ex. seedborne tip, shootborne tip)."

In the introduction, we explain that primary and lateral roots are seed-borne, whereas crown roots are shoot-borne (l. 79-80). To further clarify here, we now describe the grouping according to origin in more detail (l. 262-264).

Table S2: What is a 'seedborne tip' or 'shootborne tip'? These terms aren't defined anywhere.

We added "root" to each of the terms (e.g. "seed-borne root tip" instead of "seed-borne tip"), which should clarify our wording. Considering that we have also given further explanations in the manuscript text (previous comment), terminology is hopefully clear now.

L290 (and L263): The phrase "in dependence" is not a common usage. It keeps tripping me up, especially with phrasing like: "... in dependence on...". Is root architecture root type or root section? How are the authors defining root architecture here and elsewhere?

We replaced "in dependence on" consistently throughout the manuscript by "according to" or "related to". We use the term "root architecture" when we refer to both, root type and root section.

L291: "Domain" is the more appropriate term. Kingdom has been abandoned. (Neither are perfect, since fungi and Cercozoa are both technically in the same domain, but this is accepted as convention in microbiome research).

We respectfully disagree. The term kingdom has recently been re-defined within the domains of bacteria and archaea (doi.org/10.1099/ijsem.0.006242). However, neither kingdom nor domain are perfectly correct here. Thus, we changed it to "domain/phylum" (l. 266).

L291: What is low? Based on what information are the authors designating DESeq2 as a 'conservative' algorithm? Please provide a reference. Also, please consult: <https://doi.org/10.1093/bib/bbx104>

We compared our results to approaches that have been used in earlier studies, such as the analysis based on enrichment factor calculation (e.g. <https://doi.org/10.1016/j.soilbio.2019.03.007>), which we also applied to our data set. This enrichment factor calculation suggests the existence of many more labelled taxa and based on that calculation we considered the DeSeq analysis to be more conservative. However, reading the mentioned article, we removed the statement about the conservative DESeq2 algorithm (l. 257).

L301-309: This is a very interesting observation. I hope the authors return to this in the Discussion.

Yes, we discuss the strong labelling of fungal taxa in regions with high ^{13}C level (l. 464-492).

L309: I dislike the use of labeling the four categories: "category 1", "category 2"... Please label them based on the range of atom % ^{13}C that they are bounded by: [1.5 – 6.5), [6.5 – 11.5%), [11.5 – 16.5%), and [16.5 – 21.5%).

We consider a category denomination by using the range of atom% (or now mass fraction of ^{13}C) a bit unwieldy and prefer to keep the four category names. At the same time, we understand that this information is relevant and should not be hidden in methods and supplement. Thus, we describe the range of the mass fraction of ^{13}C % for each category now when introducing these categories in the results section (l. 259-260) and in the legend of Fig. 5. When we refer to these categories, it is mostly clear from the context whether these include samples with high or low ^{13}C level.

L310- 320: The authors should be careful in interpreting ^{13}C -enrichment among these taxa. Some of these populations could have established populations by day 15 and, thus, they appear partially labeled, when they are processing similar amounts of carbon as the highly labeled *Bacillus*, except that the *Bacillus* may have grown from a very small population size, converting more ^{13}C into DNA during replication and growth than other populations. This context is important to keep in the front of your minds, since the growth dynamics impact DNA-SIP and you are capturing only a snapshot of the temporal ('boom-bust') dynamics (see Figure 6 of this paper:

<https://www.frontiersin.org/journals/microbiology/articles/10.3389/fmicb.2016.00703/full>)

We are aware that DNA-SIP data taken at one time after the pulse provide only a snapshot of the dynamics. Consequently, we focus in our discussion on the fact that taxa were labelled (or not), but do not put too much weight on the exact label intensities or specific relative abundances, we rather look at strong contrasts.

L316: I know this will sound a bit knit-picky, but *Bacillus*, like *Paenibacillus*, are ruderal, fastgrowing organisms in part b/c of the fact they possess multiple copies of the 16S rRNA gene (on average 5, but upwards of 12 copies: <https://link.springer.com/article/10.1186/2049-2618-2-11>). You must consider this when interpreting your results.

We address this when discussing our results about *Paenibacillus* (l. 498-501).

Figure 5 – Panel C: The information shown in this panel could be summarized. I recommend aggregating the OTUs at the genus level. You can list the number of OTUs aggregated next to each name if you wish to stress this information, ex. “*Fusarium* (9)”.

An aggregation at genus level is not that easily done, because the ASVs fall into one or more different label categories. As we have color-coded the different label categories, we would have to remove it or encode this in a more complex way for ASVs within one genus, which would make the figure more difficult to understand. Thus, we would prefer to keep the figure as it is. It also nicely visualizes the few genera that are represented by different labelled ASVs. This is addressed in the discussion.

Also, please choose OTU or ASV and use it consistently throughout. This figure has a mix of both and so does the text (ex L364).

The use of ASV for bacteria and fungi and OTU for Cercozoa is on purpose. We used the highest possible meaningful resolution for each of the three groups. A presentation of Cercozoa data at ASV level is not meaningful (doi.org/10.1111/1755-0998.12729).

L375: “...condensed at phylum...” <- Try: “aggregated”

Modified as suggested (l. 312).

L334: I rarely find co-occurrence analyses a compelling exploration of ecological relationships. This analysis is no exception. What relationships do the authors believe they are mapping here?

We share the opinion that co-occurrence analyses are often overrated and we are fully aware that correlations cannot be interpreted as interactions. Despite these reservations, we see some value in this analysis here. First, it allows us to have an integrated view on Bacteria, Fungi and Cercozoa. The ^{13}C -labelling provides at least some additional evidence for potential interactions, e.g. between bacteria as prey and Cercozoa as predators. Being aware of the limitations, we emphasize some specific observations. The increasing number of correlations between Cercozoa and Bacteria along with decreasing connectedness of fungi towards root bases points to reorganization in microbial community structure and possible dependencies that change along the root axis. Further, we have a

particular focus on the ^{13}C -labeled fraction and those taxa that we identified as labeled. We assessed their particular positions within the network. The higher connectedness of these taxa at root bases aligns well with their key role as main consumers of rhizodeposits within the labeled microbial community.

L380 – 383: This observation is a plausible phenomenon, but I would like the authors to provide a limit of detection of their PET. Also, please cite the claim: “which is explained by the early developmental stage at which the root system is still supplied by seed reserves.” If this

A reference has been given to support the statement (l. 329). For more details about the detection range of our PET system, we refer to the response given to the above reviewer comment on L.116.

L404: Or, that the existing mass of ^{12}C diluted the freshly translocated photosynthate. You need to work in masses of ^{13}C instead of proportions if you’d like to make these claims.

We recalculated the data and obtained the same findings. Thus, we keep the statement (l. 351).

L412: Also, need to consider mass of ^{12}C and dilution effect of incoming label for the exact reason described: “These roots were already established with the full length that we sampled for the base section when the $^{13}\text{CO}_2$ labeling period began. In the younger roots, part of the sampled base section developed during the ^{13}C labeling period and incorporated ^{13}C -labeled photosynthates into the tissue, explaining the consistent increase in ^{13}C abundance of root samples with decreasing age.” The authors clearly understand the context, here, but need to be more cautious in other interpretations.

L418: Again, the authors did not measure ^{13}C abundance. They measured atom % ^{13}C which is a fraction. This claim, while plausible, needs to be substantiated with an analysis that accounts for the total mass of carbon in each root type.

These two points address the same problem. We have recalculated the data (see answer on comment L.157) to obtain a mass-based ^{13}C fraction in order to be able to draw conclusions on the proportion of fresh ^{13}C in root tissue and rhizosphere.

L430 – 436: This is an important point of discussion. According to Figure 2, not every root revealed by the MRI was labeled by the ^{11}C in the PET (see Figure S2 and S3)? What might account for this observation?

The figure contains scans that are “snapshots” of the labeled plants at 85-90 min. Roots that appear to be unlabeled have typically received ^{11}C before or after that imaging window. However, we chose to present this imaging timepoint as it best represented the spatial heterogeneities observed in the root system. To clarify this further, we have added panel C to Fig. S1.

L441: “When evaluating...” <- Please improve this sentence.

We rephrased the sentence (l. 389).

L449 – 451: Could you the author please explain their reasoning here. The $^{11}\text{CO}_2$ was a short pulse chase, but the influence of rhizodeposition was present prior to the temporary flush. How does this demonstrate “a comparable effect of the ^{11}C and the ^{13}C tracer on the microbiota”? It would seem that the authors link the extremely short of flush of $^{11}\text{CO}_2$ with a detectable influence on the microbiome, which is unlikely, since DNA sequencing requires that cells divide – which operates on a much longer timescale

We have deleted the statement due to the limitations mentioned by the reviewer.

L488: The authors would have a more interesting discussion of community assembly if they actually measured differences. This would be far more informative than the network analysis, and would mean this discussion was less tokenistic. Try this: <https://search.r-project.org/CRAN/refmans/iCAMP/html/bNTI.big.html>

Thank you for this valuable suggestion. We decided to dive even deeper into the analysis of assembly mechanisms using the `icamp.big()` function of the *iCamp* package. We have included the results in the supplements (Fig. S8) and describe (l. 225-236) and discuss the main findings (l. 437-461).

L499: “Towards the root basis...” <- Try: “Towards the root base, ...”

Corrected as suggested (l. 468).

L535: the noun ‘Rhizobia’ does not need to be capitalized (a genus does not have a plural form).

Corrected as suggested (l. 504).

L583: To play the devil’s advocate: your data shows a consistent ‘succession’ between younger and older roots, which reflects changes in the amount and composition of root exudates. Much of what you observed in your study matches analogous microbiome data obtained through other means. How does this suggest we need higher resolution? It suggests we need a better understanding of the shift in composition of root exudates.

We have rephrased the last section of the discussion. We have implemented the aspect about the composition of rhizodeposits, the processes that define this composition, and how this relates to changes in the microbiota. Further, we point out the important finding about the local support of taxa within the root system (l. 554-563).

L686: Do the authors mean 1.5 ug of DNA? (The use of periods to separate 1000 is confusing, since my mind reads this as significant digits of a fraction).

Corrected, it is 1.5 µg (l. 663).

704-706: Please note that it might have made more sense to pool several heavy fractions, since variation in the GC content and ¹³C-enrichment will produce variation surrounding the ‘peak’ where most 16S, ITS, or 18S sequences may be found. It isn’t necessarily a flaw, but as long as you see good separation from the background, natural abundance distribution, you can pool all heavy fractions.

Thanks for pointing this out; we will follow this suggestion in future studies in cases where the heavy fraction is well separated from the light fraction.

L751: How exactly did the authors deploy HR-SIP with their data. According to L701, the authors sequences one heavy and one light fraction. This may satisfy the minimum requirement for HRSIP, but I was under the assumption that High-Resolution SIP involved sequencing every gradient fraction. I can’t see why this method was taken (DESeq2 alone can perform the same contrast between isotopically labeled and natural abundance fractions)

As the HR-SIP function in the HTSSIP package internally employs DESeq2 for statistical modeling, we pragmatically adapted the function to focus only on key fractions. We found that the HR-SIP wrapper generally eases the interpretation of DESeq2’s output in the SIP context and will likely be comparable to future tools for metagenomic SIP analyses <https://doi.org/10.1128/msystems.01280-22>.

Data Availability: The study sequencing data was archived (BioProject: PRJNA1145186), but the sample metadata was not available on the SRA Run Selector. This could be because the authors have withheld it until publication (which is OK). Either way, please provide the metadata that will be publicly associated with your sequencing data. I was not able to evaluate whether archival was completed in a way that meets FAIR data standards and reproducibility. Furthermore, I'd like to draw the authors' attention to the new MISIP standard for guiding the archival of SIP metadata: <https://academic.oup.com/gigascience/article/doi/10.1093/gigascience/giae071/7817747>

Thank you for pointing us towards the MISIP standards, we have added the requested information to the SRA metadata file and made sure that all information is publicly available. Further raw data and scripts are now available on Github, as stated in the manuscript.

Reviewer #3 (Remarks to the Author):

Thank you very much for co-reviewing our manuscript.

Reviewer #1 (Remarks to the Author):

The authors do a good job addressing reviewer concerns regarding ¹¹C PET imaging, ¹³C SIP, and microbiome analysis. The study combines PET/MRI imaging with ¹³C labeling and microbiome community analysis and identifies spatiotemporal patterns of carbon allocation, root exudation, and microbial community assembly. These findings are likely to be of interest to the broad readership of Nature Communications. A few minor comments regarding clarity and typos are listed below.

Lines 70-73, "It is unclear...", this sentence is a bit long and confusing. It would benefit from breaking up the separate thoughts for clarity.

Divided into two sentences (l. 66).

Line 94, I think the authors mean "extends" not "extents".

Yes, corrected (l. 89).

Line 188, I think the authors are referring to seminal roots not lateral roots.

Yes, corrected (l. 187).

Line 196, I think "monotonic" is the more common form rather than "monotone".

Modified as suggested (l. 195).

Line 199, delete the period "Mortierella. increased".

Corrected as suggested (l. 198).

Line 559, delete the word "the" from the phrase "It requires the spatiotemporal to understand microbiota assembly ..."

Corrected as suggested (l. 565).

Reviewer #2 (Remarks to the Author):

Major Comments

The authors have addressed all major criticism. I have requested additional information and made suggestions to improve the clarity and impact of the article, which I hope the authors implement. The work is worthy of publication despite the semi-quantitative nature of the ¹¹C work, primarily due to the use of ¹³C-labelling paired with SIP-DNA. I plan to use some of these material in teaching my soil and rhizosphere microbiology class in Spring 2026, so I believe this will be impactful on the field.

Line Comments

L52 - 56: This is fine, but you might wish to also capture the importance of substrate concentration (i.e., a low dose of a single carbohydrate can elicit a different outcome than a high dose with the same sugar).

We agree, this plays a role as well. However, the concentration of specific compounds is not evaluated in our study and we prefer to keep this part of the introduction concise.

L93: Syntax: "Root-internal heterogeneity in C allocation extents..." This hypothesis need to be clarified.

Rephrased: "Root-internal heterogeneity in carbon allocation extends into the rhizosphere with a largely congruent pattern." (l. 88-90).

L96: The 4th hypothesis seems to be an extension of the 3rd hypothesis. It could be a separate hypothesis if the authors identify a specific microbial taxon that is selected for by a specific difference in the quality of rhizodeposit. Otherwise, it stands as a relatively obvious extension of the 3rd hypothesis.

We have rephrased the 4th hypothesis now as an independent one (l. 91-92). Even if we have not analysed the quality of rhizodeposits, we have analysed the quantity and the spatially resolved sampling along with the DNA-SIP approach allows us to address this hypothesis. We come back to this in the discussion.

L145: "at the root tips was just not significant" <- In the grand scheme of probabilities, the difference between $p = 0.049$ and $p = 0.054$ shouldn't make a major difference in your conclusions. I recommend focusing on the weaker effect (i.e., r value). Note: the symbol for Pearson's correlation coefficient is a lower case "r"

Rephrased as "correlation at the root tips was clearly weaker and not significant anymore" (l. 142-143) ; r values given as lower case r .

L197-200: Readers will want to know all OTUs that were differentially abundant across your young to old root axis. You should include this in your publication as a Supplementary Table, and also include the representative sequences for each OTU. This will help accelerate the discovery of a consensus among taxa colonizing these different root compartments.

The analysis featured in Fig. S7 was conducted at genus level, which is why no OTU IDs were extracted. We have added .FASTA files containing representative sequences for all datasets to our GitHub repository (<https://github.com/sinaschultes/photosynthate-distribution-maize-rhizosphere-microbiota>). This is mentioned in the data availability statement.

L221- 224: Nice analysis here. Good question and probably some of the best evidence that exists to make such a claim. You might wish to highlight this result in your Abstract. This part of your Abstract is a little fluffy and can be condensed into a single sentence to make room for your report of the conclusion on L221-224:

"Bacterial, fungal and protistan community structure in these rhizosphere soil samples differed depending on root structure and related spatial heterogeneities in carbon allocation. Especially ¹³C-labeled consumers of rhizodeposits, identified by DNA stable isotope probing, were responsive to photosynthate distribution. They showed differences in labelling according to their spatial localization within the root system."

We have rephrased this part of the abstract (l. 34-38).

L225-236: Your article is stronger for having completed this analysis. Nice effort.

Thanks again for this suggestion.

L237 – 252: My personal preference is to provide the alpha-diversity statistics first b/c they provide a

broader view of trends. In your case, I would recommend moving this paragraph upwards, since it dovetails nicely with what we know about alpha-diversity in the rhizosphere, namely it decreases as one moves from bulk to rhizosphere, because only a subset of taxa are most competitive / adapted to rhizosphere conditions (high C, suppressive plant secondary metabolites etc.).

We agree and usually report alpha diversity first, but did it intentionally different here. We have to begin this section by explaining how we analysed the microbiota data in the two studies to enable the reader to understand our work and subsequent data. However, alpha diversity and abundance were only evaluated in study I in detail, and it is possibly confusing to state that we have two datasets if we then present alpha diversity and abundance data for only one study. Beta diversity was a main focus in both studies and analysed comparatively across studies. Thus, we prefer to have this at the beginning of the paragraph, followed by the additional information on alpha diversity and abundance of study I.

L266: The phrase 'domain/phylum' is ambiguous here and may confuse readers, as it blends taxonomic rank with methodological grouping. Since you're referring to the different groups targeted by distinct primer sets (e.g., bacteria, fungi, Cercozoa), it would be clearer to use a term like 'amplified clade' or 'targeted taxonomic group'. Alternatively, you could rephrase the sentence to make the methodological distinction more explicit: "Across the different primer sets (bacterial 16S, fungal ITS, and cercozoan 18S), the number of identified taxa ranged from 2 to 18 per enrichment category."

We have replaced "domain/phylum" by "amplified clade" (l. 267).

L256 – 283: I recognize that you've put lots of work into further enhancing your manuscript, but I have one more important task: please be sure that all of the ASV IDs and representative sequences (FASTA) are available in your supplementary material. These are key bits of information that will enable subsequent users to map onto the work you have done. It is incredibly helpful and time saving, and will only increase the re-use of the knowledge you have gained.

FASTA files including the representative sequences of all datasets have been uploaded to the GitHub repository due to their large file size. We have added this information in the data availability section (<https://github.com/sinaschultes/photosynthate-distribution-maize-rhizosphere-microbiota>).

Figure 5: Use consistent labels, re: "OTU" or "ASV"

As explained in the previous revision, we have analysed bacteria and fungi at ASV level, but Cercozoa at OTU, level to have the highest meaningful taxonomic resolution for each group. Thus, it would be incorrect to modify the label.

L437 – 461: I agree with these new conclusions.

L472 - 478: Sure, but many Fusarium are also beneficial to hosts. In general, the pathogens are cheaters who co-opt the trust gained by the beneficials. Fusarium is a highly diverse genus that includes not only notorious pathogens (e.g., *F. oxysporum*, *F. graminearum*) but also many non-pathogenic endophytes and even plant growth-promoting strains, including strain level variation (re: <https://www.frontiersin.org/journals/microbiology/articles/10.3389/fmicb.2015.01248/full>). I recommend more nuance here. A subjective focus on pathogens is an age-old mistake when it comes to host-microbe interactions.

We agree and have rephrased our statements to better account for the existence of commensal and beneficial *Fusarium* strains in maize (l. 482-483).

L492 – 495: Before concluding this, please verify that your fungal primer set has been validated for capturing AMF. The Glomeromycota are sometimes missed by different ITS sets. The Tedersoo (2017) paper (“PacBio metabarcoding of Fungi and other eukaryotes: errors, biases and perspectives”) might be able to assist in this effort.

The primer pair we applied has been used in studies focussing on AMF (e.g. Gil-Fernández et al 2025, *Mycorrhiza*, doi.org/10.1007/s00572-025-01210-x), but may not be the best to detect AMF (Schlaeppli et al 2016, *New Phytologist*, doi.org/10.1111/nph.14070). We may have overlooked AMF in case they were present at very low relative abundance. As we cannot exclude this and as the statement about AMF is not important to support conclusions of this article, we deleted the statement rather than elaborating on possible methodological limitations.

L514 – 532: Are there ways that Cercozoa may be labeled directly by exudate uptake? Do they ingest solution when they gulp bacteria? I think your conclusion about predation is very interesting and impactful, so if you could strengthen it, then I recommend mentioning it in your abstract. This is the kind of ecological insight that your high-resolution C tracking can provide direct evidence of.

The great majority of Cercozoa in the rhizosphere are raptorial feeding flagellates (*Cercomonas*, *Glissomonas*), which means that they use one of their flagella to actively hunt for bacteria. Accordingly, we consider them bacterivores. As suggested, we emphasize the predatory aspect more explicitly in the abstract, which covers besides the Cercozoa also predatory bacteria in our study (l. 40-41).

General comments

The study by Schultes *et al.* provides a high resolution view into the spatial allocation of photosynthate to roots and rhizodeposits, and its structuring of the rhizomicrobiome. Their methodology provided direct measures of the translocation to root hotspots, and their observations are high quality, impactful, and internally consistent. While I found the experiment elegant, I did feel that the hypotheses were fairly generic, so much so that the authors did not bother to explicitly revisit them in their Discussion. As an exploration of hard-to-observe, and poorly described, belowground phenomena, that may be forgivable. The authors should be lauded for studying across domains (scales) of soil fauna and to pair and contrast to two isotope tracing methods.

Overall, the analysis and interpretation of results are valid (though I have many clarifying questions in the line comments). However, there is one major exception: the authors refer to ‘¹³C abundance’ in the text but the measurements they obtained are atom % ¹³C, which is a proportion of ¹³C relative to the total C. The conflation of proportion ¹³C with abundance may impact several of the authors’ conclusions and **will need to be addressed**. Accounting for differences in the mass of root (from which the proportion is measured) will be necessary to draw several conclusions made by the authors. While this is a current flaw in their analysis and interpretation, it can be corrected. In fact, I urge the authors to try to provide quantitative estimates, where possible, for the mass of photosynthate accumulating among these different root types. In general, the authors default to generalizations of their measures of intensity. Could they explain why they did not attempt to make estimates of abundance? Admittedly, these measure could be biased by assumptions, but given the MRI data and the capacity to calculate root masses, this could (in theory) be done.

In hindsight, the work would have been more impactful if paired with shotgun metagenomic sequencing, since pairing hotspots of rhizodeposition with functional activity would have been very revealing. Yet, this is forgivable, since the effort and expense were substantial. Lastly, there were some missing references that could be particularly useful here: [10.1038/s42003-021-01988-4](https://doi.org/10.1038/s42003-021-01988-4), [10.1016/j.soilbio.2019.05.011](https://doi.org/10.1016/j.soilbio.2019.05.011), [10.1038/s41564-018-0129-3](https://doi.org/10.1038/s41564-018-0129-3), and <https://pubmed.ncbi.nlm.nih.gov/36964135/> (and listed in the **Line Comments**).

Line comments

L32: Syntax. Do the authors mean: “...in *the* rhizosphere microbiome”?

L43: Rhizodeposit indicates that the communities were from mineral particles, is that correct?

L45: “...profited most from rhizodeposits in specific root regions.” <- This sentence is good, but could you be more specific? Maybe: “...profited most from rhizodeposits in specific root regions *where rhizodeposition was greatest*” or some other description.

L46: The term ‘niche differentiation’ connotes a specific ecological process, which focuses on how competing organisms will adapt (‘differentiate’) to a new niche to lessen competition. I do

not think this is what the authors have in mind here (I could be wrong). If I'm correct, please use a different choice of words.

L65: Please clarify 'this'. The variation in root types and structure? The current meaning suggests 'Complex root systems likely result in the development of root-type specific microbiomes.' I don't completely follow the logic here: "because roots systems are complex, there should be root-type specific microbiomes." It would be stronger if you argued: 'different root types produce different rhizodeposits (ref). And, root systems are comprised of various root types and architectures (ref). Thus, root systems will have a root-type specific microbiomes.'

L69-72: While I am interested in the system you are studying, the justification of the system remains "largely unresolved at the small scale" is not a compelling one. What might we better understand about the soil / rhizosphere microbiome or plant-mediated microbial activity in soil by resolving the spatial variation in the rhizosphere microbiome? In other words, one can argue that we haven't resolved fine-scale dynamics in many environmental microbiomes, why this one? I don't think the authors will have a hard time arguing why, but I'd like them to present a strong justification.

L86: Syntax: "...and (III), in particular, microbial consumers..."

L83 – 86: Similar to the justifications provided, the hypotheses are basic. The authors are doing something never been done before, but the justifications ("it's never been done") and hypotheses ("we'll see heterogeneity, we'll see growth patterns in the rhizosphere that map to where the carbon is flowing, and microbes that feed on rhizodeposits will benefit most"). The author could provide a more compelling reason for why their high-resolution technique is valuable. For starters, it gives us greater certainty over who is responding to fresh rhizodeposits. One could, in theory, chart diurnal and developmental influences of plant on soil / rhizosphere biology. It's clearly exciting, but the hypotheses leave much to be desired.

L87: The belowground distribution of photosynthate to be spatially and temporally dynamic and that these patterns would produce variation in rhizosphere extent and activity. A review from 5 years ago make a similar case: [10.1016/j.soilbio.2019.05.011](https://doi.org/10.1016/j.soilbio.2019.05.011). Here are some primary sources: [10.1038/s41564-018-0129-3](https://doi.org/10.1038/s41564-018-0129-3), <https://pubmed.ncbi.nlm.nih.gov/36964135/>. All three are pretty key papers for the topic of the present work.

L90 (also, L112, L138 & L607): How did the authors decide on their sampling schedule of 6, 13, and 21 days?

L94 (also missing from section on L618): How where the photosynthate levels determined?

L112: It might be better to call these 'growth stages' or 'developmental stages', since they correspond to the size of the root system. This will help differentiate between the temporally-resolved data you present in Figure 2b / the video.

L113: What is the sensitivity of the ¹¹C PET scan? Your observation is valid in relative terms, but you need to acknowledge your level of detection.

L116: Did the authors try to estimate how much C was present in the ‘high,’ ‘medium’, and ‘low’ photosynthate classes using the ^{13}C data? Do the authors know whether the ‘high’ signal is saturating the detector? I am not an expert in PET, so the manuscript would be more informative if the authors helped me understand how quantitative the signal is and what the limitations are. Put simply, what is the range between 0 and ‘max’ for your detector?

L132: Can the authors quantify how many of the roots did not receive any ^{11}C label relative to the total root mass? The temporal dynamic is interesting, but so is the observation that some roots do not appear to channel freshly fixed ^{11}C .

L157: The authors refer to ‘ ^{13}C abundance’ in the text but the measurements they obtained are **atom % ^{13}C** , which is a proportion of ^{13}C relative to the total C (i.e., the remainder being ^{12}C). Thus, the terms ‘relative abundance’, ‘proportion’, or ‘enrichment’ are more appropriate.

The authors should also report total ^{13}C or even ‘excess ^{12}C equivalent’ normalized to the C mass in their plant tissues (see: <https://esajournals.onlinelibrary.wiley.com/doi/full/10.1890/08-1884.1> for calculations). The mass of each root type will impact their conclusions, given the new influx of fresh ^{13}C label will be diluted to differing degrees proportional to the existing ^{12}C mass. The author should try to estimate root masses for each root type based on the root volume (MRI) using assumptions about the mass and quantity of C in root tissues. For root tissues, this could be tractable, while the rhizosphere might get a little squishy. The authors should explore this in order to gain a better understanding of the carbon dynamics. Arguably, there will be error biases that result from estimations, but this should be at least considered and presented along with the atom % ^{13}C measurements.

L168: “For the tissue of the root tips, the ^{13}C abundance increased with decreasing root age, ...”
<- Two thoughts: (i) can the authors write more clearly with less jargon? “decreasing root age” = “younger” and (ii) younger roots will have received more label by default, since the new roots will be comprised of recently fixed C (in your case $^{13}\text{CO}_2$). The value of these observations is limited in my opinion, especially without adequate context about what the authors expected to observe related to their hypotheses.

L168: Could this discrepancy between base root (with a relatively consistent mass) and root tips (with a much more variable mass) play a role in the variation you observe. As in, if you normalize to root mass, will the correlation between enrichment and rhizosphere in root tips strengthen?

L171: “The decline in...” <- This sentence could be more clearly articulated. Too many conditional statements and too long.

L202: “...differed in dependence...” <- awkward wording. Try: “...differed according to...”

L203: PERMANOVA does not really ‘validate’ results – it provides evidence. The authors have done a nice job with their statistical analyses, but some of the wording could be refined.

L211: “...but a certain successiveness was also evident.” <- This is too vague. Please describe the phenomenon. Please do not make me imagine what ‘a certain successiveness’ means.

L212: This is repetitive. PERMANOVA is not a measure of sequential change. You might wish to try a Procrustes analysis for this (T1 -> T2 vs. T1 -> all others, then T2 -> T3 vs. T2 -> all others etc.). If the succession from T1->T2 is a better fit, for example, you have grounds to describe a form of succession.

L216-220: These are nice observations, but it would be best to back up your claims about ‘sample clustering ...reflecting the chronology of root emergence.’

L223 - 228: This is strong evidence communicated effectively. Nice job. It is a nice demonstration that your methods worked for capturing variations in the hotspots of rhizosphere activity.

L285 – 289: The description is not clear. I believe I understand what the authors did, but it took some head scratching. Please clarify.

L288: The authors have used “root type” and “root section” consistently up until now. What is ‘the origin in the root system’? It might be useful to give examples in a bracket, like: “Alternatively, samples were grouped according to their origin in the root system (*ex.* seedborne tip, shootborne tip).”

Table S2: What is a ‘seedborne tip’ or ‘shootborne tip’? These terms aren’t defined anywhere.

L290 (and L263): The phrase “in dependence” is not a common usage . It keeps tripping me up, especially with phrasing like: “... in dependence on...”. Is root architecture root type or root section? How are the authors defining root architecture here and elsewhere?

L291: “Domain” is the more appropriate term. Kingdom has been abandoned. (Neither are perfect, since fungi and Cercozoa are both technically in the same domain, but this is accepted as convention in microbiome research).

L291: What is low? Based on what information are the authors designating DESeq2 as a ‘conservative’ algorithm? Please provide a reference. Also, please consult: <https://doi.org/10.1093/bib/bbx104>

L301-309: This is a very interesting observation. I hope the authors return to this in the Discussion.

L309: I dislike the use of labeling the four categories: “category 1”, “category 2”... Please label them based on the range of atom % ¹³C that they are bounded by: [1.5 – 6.5), [6.5 – 11.5%), [11.5 – 16.5%), and [16.5 – 21.5%).

L310- 320: The authors should be careful in interpreting ¹³C-enrichment among these taxa. Some of these populations could have established populations by day 15 and, thus, they appear partially labeled, when they are processing similar amounts of carbon as the highly labeled *Bacillus*, except that the *Bacillus* may have grown from a very small population size, converting more ¹³C into DNA during replication and growth than other populations. This context is important to keep in the front of your minds, since the growth dynamics impact DNA-SIP and you are capturing only a snapshot of the temporal (‘boom-bust’) dynamics (see Figure 6 of this paper: <https://www.frontiersin.org/journals/microbiology/articles/10.3389/fmicb.2016.00703/full>)

L316: I know this will sound a bit knit-picky, but *Bacillus*, like *Paenibacillus*, are ruderal, fast-growing organisms in part b/c of the fact they possess multiple copies of the 16S rRNA gene (on average 5, but upwards of 12 copies: <https://link.springer.com/article/10.1186/2049-2618-2-11>). You must consider this when interpreting your results.

Figure 5 – Panel C: The information shown in this panel could be summarized. I recommend aggregating the OTUs at the genus level. You can list the number of OTUs aggregated next to each name if you wish to stress this information, ex. “*Fusarium* (9)”. Also, please choose OTU or ASV and use it consistently throughout. This figure has a mix of both and so does the text (ex L364).

L375: “...condensed at phylum...” <- Try: “aggregated”

L334: I rarely find co-occurrence analyses a compelling exploration of ecological relationships. This analysis is no exception. What relationships do the authors believe they are mapping here?

L380 – 383: This observation is a plausible phenomenon, but I would like the authors to provide a limit of detection of their PET. Also, please cite the claim: “which is explained by the early developmental stage at which the root system is still supplied by seed reserves.” If this

L404: Or, that the existing mass of ^{12}C diluted the freshly translocated photosynthate. You need to work in masses of ^{13}C instead of proportions if you’d like to make these claims.

L412: Also, need to consider mass of ^{12}C and dilution effect of incoming label for the exact reason described: “These roots were already established with the full length that we sampled for the base section when the $^{13}\text{CO}_2$ labeling period began. In the younger roots, part of the sampled base section developed during the ^{13}C labeling period and incorporated ^{13}C -labeled photosynthates into the tissue, explaining the consistent increase in ^{13}C abundance of root samples with decreasing age.” The authors clearly understand the context, here, but need to be more cautious in other interpretations.

L418: Again, the authors did not measure ^{13}C abundance. They measured atom % ^{13}C which is a fraction. This claim, while plausible, needs to be substantiated with an analysis that accounts for the total mass of carbon in each root type.

L430 – 436: This is an important point of discussion. According to Figure 2, not every root revealed by the MRI was labeled by the ^{11}C in the PET (see Figure S2 and S3)? What might account for this observation?

L441: “When evaluating...” <- Please improve this sentence.

L449 – 451: Could you the author please explain their reasoning here. The $^{11}\text{CO}_2$ was a short pulse chase, but the influence of rhizodeposition was present prior to the temporary flush. How does this demonstrate “a comparable effect of the ^{11}C and the ^{13}C tracer on the microbiota”? It would seem that the authors link the extremely short of flush of $^{11}\text{CO}_2$ with a detectable influence on the microbiome, which is unlikely, since DNA sequencing requires that cells divide – which operates on a much longer timescale

L488: The authors would have a more interesting discussion of community assembly if they actually measured differences. This would be far more informative than the network analysis, and would mean this discussion was less tokenistic. Try this: <https://search.r-project.org/CRAN/refmans/iCAMP/html/bNTI.big.html>

L499: “Towards the root basis...” <- Try: “Towards the root base, ...”

L535: the noun ‘Rhizobia’ does not need to be capitalized (a genus does not have a plural form).

L583: To play the devil’s advocate: your data shows a consistent ‘succession’ between younger and older roots, which reflects changes in the amount and composition of root exudates. Much of what you observed in your study matches analogous microbiome data obtained through other means. How does this suggest we need higher resolution? It suggests we need a better understanding of the shift in composition of root exudates.

L686: Do the authors mean 1.5 ug of DNA? (The use of periods to separate 1000 is confusing, since my mind reads this as significant digits of a fraction).

704-706: Please note that it might have made more sense to pool several heavy fractions, since variation in the GC content and ¹³C-enrichment will produce variation surrounding the ‘peak’ where most 16S, ITS, or 18S sequences may be found. It isn’t necessarily a flaw, but as long as you see good separation from the background, natural abundance distribution, you can pool all heavy fractions.

L751: How exactly did the authors deploy HR-SIP with their data. According to L701, the authors sequences one heavy and one light fraction. This may satisfy the minimum requirement for HR-SIP, but I was under the assumption that High-Resolution SIP involved sequencing every gradient fraction. I can’t see why this method was taken (DESeq2 alone can perform the same contrast between isotopically labeled and natural abundance fractions)

Data Availability: The study sequencing data was archived (BioProject: PRJNA1145186), but the sample metadata was not available on the SRA Run Selector. This could be because the authors have withheld it until publication (which is OK). Either way, please provide the metadata that will be publicly associated with your sequencing data. I was not able to evaluate whether archival was completed in a way that meets FAIR data standards and reproducibility. Furthermore, I’d like to draw the authors’ attention to the new MISIP standard for guiding the archival of SIP metadata: <https://academic.oup.com/gigascience/article/doi/10.1093/gigascience/giae071/7817747>